# RNA stability controlled by m⁶A methylation contributes to X-to-autosome dosage compensation in mammals

Cornelia Rücklé[1,6], Nadine Körtel[1,6], M. Felicia Basilicata [1,2], Anke Busch [1], You Zhou [3], Peter Hoch-Kraft[1], Kerstin Tretow[1], Fridolin Kielisch [1], Marco Bertin[2], Mihika Pradhan[1], Michael Musheev [1], Susann Schweiger [1,2], Christof Niehrs [1,4], Oliver Rausch [5], Kathi Zarnack [3], Claudia Isabelle Keller Valsecchi [1] & Julian König [1]✉

In mammals, X-chromosomal genes are expressed from a single copy since males (XY) possess a single X chromosome, while females (XX) undergo X inactivation. To compensate for this reduction in dosage compared with two active copies of autosomes, it has been proposed that genes from the active X chromosome exhibit dosage compensation. However, the existence and mechanisms of X-to-autosome dosage compensation are still under debate. Here we show that X-chromosomal transcripts have fewer m⁶A modifications and are more stable than their autosomal counterparts. Acute depletion of m⁶A selectively stabilizes autosomal transcripts, resulting in perturbed dosage compensation in mouse embryonic stem cells. We propose that higher stability of X-chromosomal transcripts is directed by lower levels of m⁶A, indicating that mammalian dosage compensation is partly regulated by epitranscriptomic RNA modifications.

Sex chromosomes evolved from a pair of autosomes. During this process, the chromosome present in only the heterogametic sex (that is, the Y chromosome in male mammals) acquires mutations, undergoes recurrent chromosomal rearrangements, and eventually becomes highly degenerated, gene-poor, and heterochromatic[1]. Consequently, the X chromosome and most of its genes are present in a single copy in males, whereas two X chromosomes are present in females. To equalize expression between sexes in eutherian female mammals, one randomly chosen X chromosome is inactivated ($X_i$) early in development at around the implantation stage, while the other X chromosome remains active ($X_a$). Therefore, XY males and $X_iX_a$ females exhibit an imbalance of gene dosage between sex chromosomes and autosomes, which are present in one and two active copies, respectively[2]. To restore the balance between X chromosomes and autosomes, Susumu Ohno

hypothesized that the expression of X-chromosomal genes is upregulated by twofold[3]. Indeed, there are several mechanisms for how this could be achieved. For instance, previous studies have proposed that higher RNA polymerase II occupancy, as well as more activating epigenetic marks and gains in chromatin accessibility on the X chromosome, plays a role in dosage compensation[4–7]. Additionally, higher RNA stability of X-chromosomal transcripts has been observed[6,8]. There is evidence that nonsense-mediated mRNA decay (NMD) targets are enriched for autosomal transcripts[9], which could partially explain the higher RNA stability of X-chromosomal transcripts. Another recent study has proposed that dosage compensation could be mediated by elevated translation of X-chromosomal transcripts[10]. Eventually, dosage compensation may be required for only a certain subset of transcripts that are dosage-sensitive, for instance, if stoichiometry with

[1]Institute of Molecular Biology (IMB), Mainz, Germany. [2]Institute of Human Genetics, University Medical Center of the Johannes Gutenberg University Mainz, Mainz, Germany. [3]Buchmann Institute for Molecular Life Sciences (BMLS) & Institute of Molecular Biosciences, Goethe University Frankfurt, Frankfurt, Germany. [4]Division of Molecular Embryology, DKFZ-ZMBH Alliance, Heidelberg, Germany. [5]STORM Therapeutics Ltd., Cambridge, UK. [6]These authors contributed equally: Cornelia Rücklé, Nadine Körtel. ✉e-mail: j.koenig@imb-mainz.de

transcripts from other chromosomes is necessary for proper complex formation[11]. Some dosage-sensitive transcripts may also be protected from the degeneration process occurring on the Y chromosome and thus be retained in two copies[12]. However, Ohno's hypothesis is still under investigation, and both transcriptional and post-transcriptional mechanisms could play a role or act together[10,13–17]. If the latter is the case, this creates the conundrum of how the chromosomal origin of a transcript is 'remembered' in downstream steps of gene expression that occur at the RNA level.

RNA modifications are increasingly being recognized for their role in post-transcriptional gene regulation. By their 'epitranscriptomic' nature, they have the potential to bridge DNA context to mRNA fate. $N^6$-methyladenosine (m⁶A) is the most abundant internal mRNA modification, with estimates ranging from 1 to 13 modifications present per transcript[18–21]. Conserved adenine methyltransferases, such as METTL3, co-transcriptionally modify nascent mRNAs in the nucleus. The majority of m⁶A sites occur within a DRACH motif (that is, [G/A/U][G>A]m⁶AC[U>A>C]), with GGACH as the predominantly methylated sequence[22–24]. m⁶A-methylated transcripts recruit different reader proteins. Most prominently, YTHDF proteins (YTHDF1, YTHDF2, and YTHDF3) reduce the stability of m⁶A-modified transcripts in the cytoplasm by promoting their degradation[25–27]. Hence, m⁶A modifications affect mRNA fate in the cytoplasm upon their deposition in the nucleus.

In this Article, we show that m⁶A RNA modifications play a key role in X-to-autosome dosage compensation. We find that the m⁶A content is reduced in transcripts from the X chromosome, leading to more stable transcripts and longer half-lives. This is crucial to equalize the imbalance in gene dosage between autosomes and the X chromosome.

## Results

### Autosomal transcripts are stabilized by m⁶A depletion

One of the most prominent functions of m⁶A is regulating mRNA levels by promoting RNA decay[25]. It has been proposed that X-chromosomal transcripts are more stable than autosomal transcripts[6,8], so we hypothesized that m⁶A-mediated RNA stability may be involved in X-to-autosome dosage compensation. To investigate this, we first confirmed the chromosomal differences in RNA stability in published mRNA half-lives from mouse embryonic stem cells (mESCs), measured by thiol(SH)-linked alkylation for the metabolic sequencing of RNA (SLAM-seq)[28]. Indeed, transcripts originating from the X chromosome had significantly longer half-lives than autosomal transcripts (Extended Data Fig. 1a).

To investigate the direct impact of m⁶A depletion, we employed the small-molecule inhibitor STM2457, which specifically targets the major mRNA m⁶A methyltransferase Mettl3 (ref. 29). We corroborated in a time-course experiment that the m⁶A levels already showed a strong reduction after 3 h and reached a low point after 6 h of inhibitor treatment (Extended Data Fig. 1b). Compared with a *Mettl3* knockout (KO), this acute m⁶A depletion enabled us to investigate the immediate response to m⁶A depletion while minimizing secondary effects[30]. Expression analysis of marker genes[31] and quantitative polymerase chain reaction (qPCR) validations showed that the pluripotent state of the mESC remained unimpaired throughout the treatment (Extended Data Fig. 1c,d).

To determine the effect of m⁶A depletion on mRNA half-lives, we performed SLAM-seq in m⁶A-depleted and control conditions (6 h STM2457-treated or DMSO-treated as control, Fig. 1a and Extended Data Fig. 2a,b). We achieved a stable 4-thiouridine (s⁴U) incorporation rate of 1.36% after 24 h of labeling, which gradually decreased upon washout (Extended Data Fig. 2c). By fitting the SLAM-seq data using an exponential decay model and filtering for expression and a sufficient goodness of fit (see Methods)[28], we obtained half-life estimates for 7,310 transcripts (Supplementary Table 1, Fig. 1b,c, and Extended Data Fig. 2d,e). The estimated half-lives in the control condition correlated well with previously published mRNA half-lives[28] (Extended Data Fig. 2f).

Consistent with a role for m⁶A in destabilizing transcripts[25,32], the median half-life of mRNAs significantly increased upon acute

m⁶A depletion (Fig. 1b,c). Using high-confidence m⁶A sites, which we had previously mapped in the same cell line using miCLIP2 (m⁶A individual-nucleotide resolution ultraviolet (UV) crosslinking and immunoprecipitation) and m⁶Aboost[33], we confirmed that, in control conditions, transcripts with m⁶A sites had significantly shorter half-lives than did unmethylated transcripts[28] (Fig. 1d). Furthermore, the transcripts with m⁶A sites were significantly stabilized upon acute m⁶A depletion (8% median increase), whereas unmethylated transcripts were largely unaffected (0.3% median decrease, Fig. 1e).

Having ensured the high quality of our dataset, we turned to chromosomal differences in mRNA stability. X-chromosomal transcripts had significantly longer half-lives than autosomal transcripts under control conditions (Extended Data Fig. 2g, left). Importantly, the half-lives of autosomal transcripts significantly increased after acute m⁶A depletion (5% median increase), whereas the stability of X-chromosomal transcripts remained unchanged (0.2% median decrease, Fig. 1f). Transcripts on all autosomes responded similarly, while the X chromosome was the only chromosome that seemed to be excluded from this increase (Fig. 1g and Extended Data Fig. 2g). These results indicated that m⁶A-mediated RNA stability could play a direct role in X-to-autosome dosage compensation in mESCs. To further support this, we reanalyzed published mRNA half-lives for wild-type (WT) and *Mettl3* KO mESCs[34] and observed the same difference in RNA stabilization between X-chromosomal and autosomal transcripts (Fig. 1h). The deviation in absolute values between the two experiments may result from chromosomal differences or from compensatory mechanisms after KO generation, such as induced expression of alternatively spliced *Mettl3* isoforms[30]. Collectively, the intersection between our experiments and published data conclusively shows that m⁶A modifications destabilize autosomal transcripts, while X-chromosomal transcripts are largely excluded from such regulation.

### X-chromosomal transcripts are less affected by m⁶A depletion

To test whether the chromosomal differences in RNA stability contribute to balancing expression levels between the X chromosome and autosomes, we performed RNA sequencing (RNA-seq) experiments to measure the transcript expression levels after m⁶A depletion (24 h STM2457, Extended Data Fig. 3a and Supplementary Table 2). The degree of upregulation correlated with the number of m⁶A sites, such that the most heavily methylated transcripts showed the strongest upregulation (Extended Data Fig. 3c). Strikingly, we observed a marked difference in the response to m⁶A depletion between X-chromosomal and autosomal transcripts. On autosomes, we found more upregulated genes relative to the X chromosome, whereas the X-chromosomal transcripts showed by far the lowest median fold change of all chromosomes (Fig. 2a). Between autosomes, observed changes were very similar, suggesting that transcripts on all autosomes were equally affected by acute m⁶A depletion.

To directly assess the balance between X-chromosomal and autosomal transcript levels, we determined the X-chromosomal-to-autosomal (X:A) expression ratio[5,35]. In DMSO-treated cells, the median X:A ratio was approximately one when excluding silent genes or those with low expression, illustrating that X-to-autosome dosage compensation is functional in male mESCs (Extended Data Fig. 3d,e). Importantly, the X:A ratio significantly decreased in the m⁶A-depleted conditions, indicating that m⁶A depletion leads to an imbalance in X-to-autosome dosage compensation (Fig. 2b). We note that the X:A ratio does not reach 0.5, suggesting that m⁶A acts in addition to other regulatory mechanisms in X-to-autosome dosage compensation.

The differential effects of m⁶A depletion on X-chromosomal and autosomal genes were further supported in a time-course RNA-seq experiment with 3–12 h STM2457 treatment (Extended Data Fig. 1b,c and Supplementary Table 2). Of note, autosomal transcripts showed a distinct response from X-chromosomal transcripts already after 6 h of m⁶A depletion, which persisted throughout 9 h and 12 h of treatment

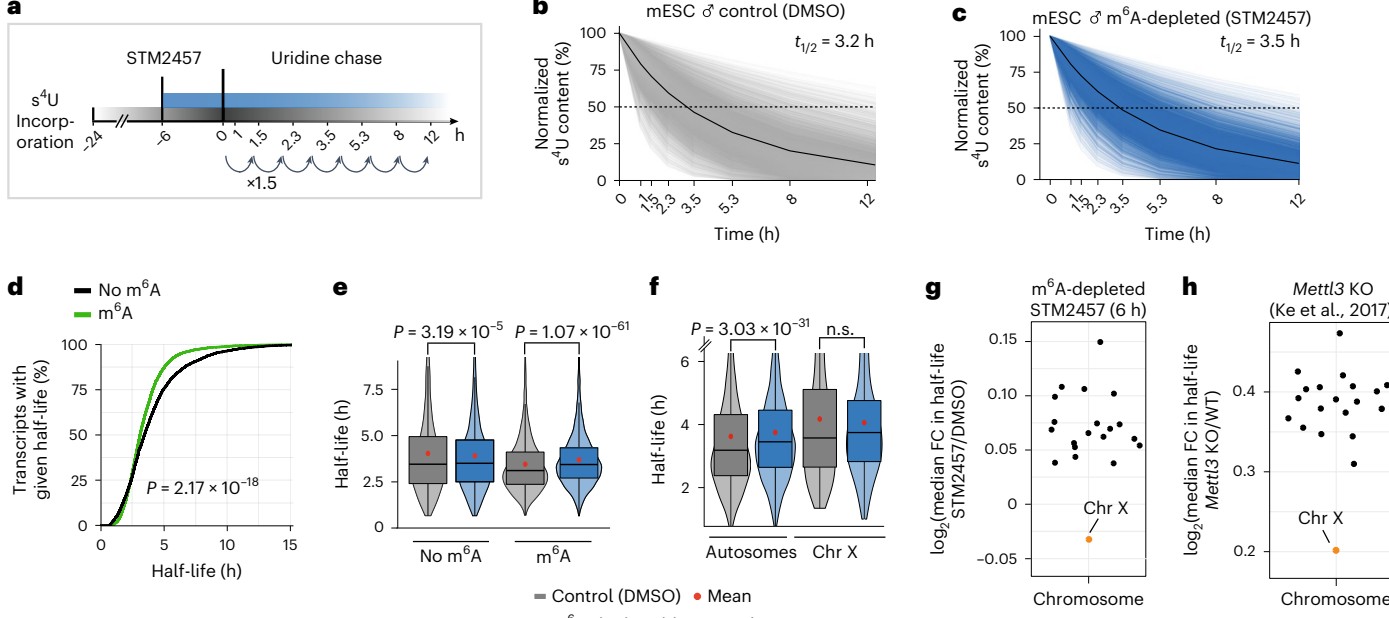

**Fig. 1 | X-chromosomal transcripts are more stable upon m⁶A depletion.**
**a**, Experimental setup for the SLAM-seq experiment. **b,c**, Transcripts ($n = 7{,}310$) in control (**b**) and m⁶A-depleted conditions (**c**) show a median half-life ($t_{1/2}$) of 3.2 h and 3.5 h, respectively ($P$ value $= 5.25 \times 10^{-29}$, two-tailed Wilcoxon signed-rank test). Median s⁴U content for all transcripts is shown in black. **d**, Transcripts with m⁶A sites have significantly shorter half-lives ($P$ value $= 2.17 \times 10^{-18}$, two-tailed Wilcoxon rank-sum test). Cumulative fractions of transcripts with given half-lives for transcripts with ($n = 2{,}342$, green) or without ($n = 4{,}967$, black) m⁶A sites. **e**, Transcripts with m⁶A sites ($n = 2{,}342$) significantly increase in half-life upon m⁶A depletion (8% median increase, $P$ value $= 1.07 \times 10^{-61}$, two-tailed Wilcoxon signed-rank test), unmethylated transcripts ($n = 4{,}967$) were largely unaffected (0.3% median decrease, $P$ value $= 3.186 \times 10^{-5}$) (same gene set in both conditions). The mean half-life in each group is shown as a red dot. Boxes represent quartiles, center lines denote medians, and whiskers extend to most extreme values within $1.5 \times$ interquartile range. **f**, Half-lives of autosomal transcripts significantly increase upon m⁶A depletion ($P$ value $= 3.03 \times 10^{-31}$, two-tailed Wilcoxon signed-

rank test), while X-chromosomal transcripts remain unchanged ($P$ value $= 0.2121$, two-tailed Wilcoxon signed-rank test). Distribution of half-lives for autosomal ($n = 7{,}069$) and X-chromosomal transcripts ($n = 241$) (same gene set in both conditions). The mean half-life in each group is shown as a red dot. Boxes are as in **e**. **g**, Median fold change (FC) in mRNA half-lives (log₂) for each chromosome in m⁶A-depleted (STM2457) over control (DMSO) conditions. X-chromosomal transcripts show the lowest half-life increase upon m⁶A depletion ($P$ value $= 0.005486$, mean difference in log₂(fold change) values $= -0.0945$, linear mixed model, two-tailed $t$-test of fixed effects, see Methods). **h**, Median fold change (log₂) in mRNA half-lives for each chromosome in *Mettl3* KO over WT mESCs[34] ($P$ value $= 0.000225$, X-chromosomal versus autosomal transcripts, mean difference in log₂-transformed fold changes $= -0.22057$). The absolute differences between m⁶A depletion and *Mettl3* KO conditions may result from differences in the experimental setup, including the mode of Mettl3 inactivation and the method used to determine transcript half-lives.

---

(Fig. 2c and Extended Data Fig. 4a, b). This was validated by qPCR for five autosomal and five X-chromosomal transcripts after 9 h of m⁶A depletion (Extended Data Fig. 4c). The clear separation of X-chromosomal and autosomal transcripts at around 6 h was in line with the observed mRNA stability changes after the same treatment duration (Fig. 1g) and supported a direct effect of m⁶A in transcript destabilization.

Next, we investigated whether m⁶A similarly regulates X-chromosomal transcripts in humans. To this end, we performed RNA-seq of primary human fibroblasts (male) after 9 h of m⁶A depletion (Fig. 2d and Extended Data Fig. 5a). As in mESCs, we observed a clear separation of the X chromosome and autosomes, such that X-chromosomal transcripts displayed significantly lower expression changes in response to m⁶A depletion (Fig. 2d). This was further corroborated by RNA-seq data colleted following m⁶A depletion in human HEK293T (female), C643 (male), and RPE1 (female) cells, which consistently demonstrated the same effect across all cell types (Extended Data Fig. 5a,b). Similar to our findings in mESCs, X:A expression ratios were close to one for human fibroblasts and RPE1 cells, whereas higher median X:A ratios were obtained for HEK293T and C643 cells, possibly owing to aneuploidies (Fig. 2e). Importantly, the X:A ratio was significantly lowered in all cases in response to m⁶A depletion, indicating that m⁶A depletion results in an imbalance of X-chromosomal to autosomal transcript expression. We conclude that the same mechanism we observe in mice is also active in humans, whereby autosomal and

X-chromosomal transcripts are differentially affected by m⁶A depletion. Our data thus support a conserved role for m⁶A in X-to-autosome dosage compensation in mammals.

## m⁶A is reduced on transcripts from the X chromosome
Our RNA-seq data showed that autosomal transcripts are more susceptible to m⁶A depletion than are X-chromosomal transcripts. To test whether these differences are driven by different methylation levels, we analyzed the distribution of m⁶A sites across chromosomes in male mESCs using miCLIP2 data[33]. Because m⁶A detection in miCLIP2 experiments partially depends on the underlying RNA abundance[33], we quantified m⁶A sites within expression bins (Extended Data Fig. 6a). Remarkably, 74.5% of all transcripts with intermediate expression (bins 4–8) harbored at least one m⁶A site, with an average of one to five m⁶A sites per transcript. By contrast, on transcripts with low expression (bins 1–3), we found no m⁶A sites in most cases, most likely owing to detection limits (Fig. 3a and Extended Data Fig. 6b).

Intriguingly, separation by chromosomes revealed a significantly lower level of m⁶A modifications on X-chromosomal transcripts, which were reduced by almost half compared with the genomic average (56% remaining, Fig. 3b). By contrast, transcripts on all autosomes showed similar numbers of m⁶A sites (Fig. 3c and Extended Data Fig. 6c). For further quantification, we calculated the average fold change in m⁶A numbers on a given chromosome relative to all chromosomes.

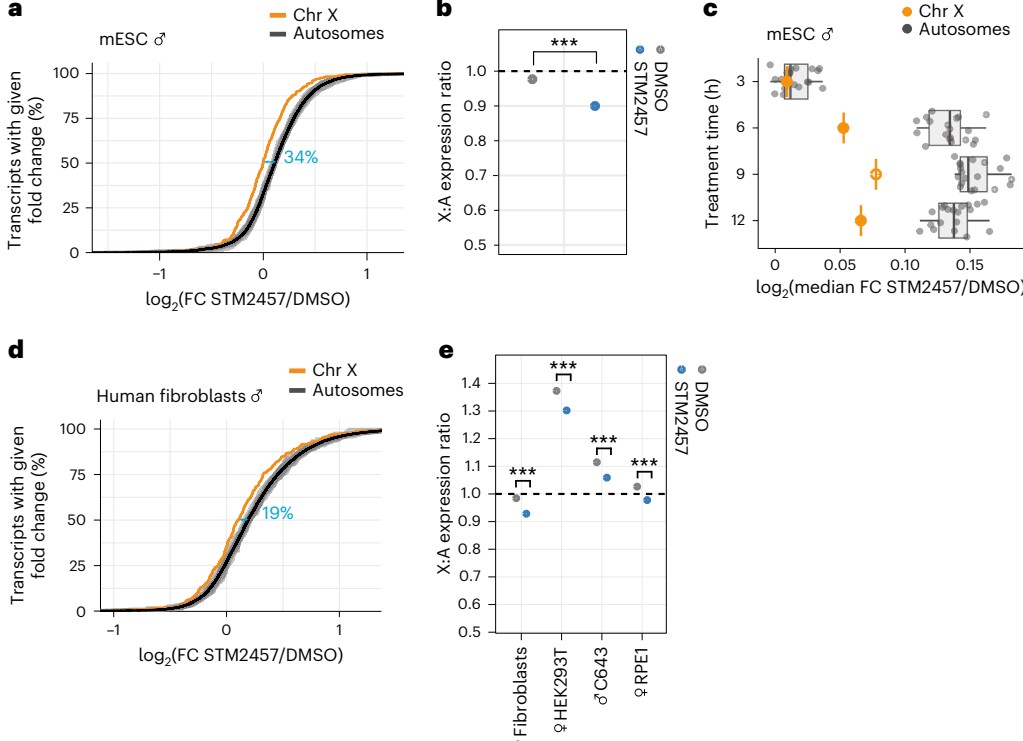

**Fig. 2 | X-chromosomal transcripts are more stable and less upregulated upon m⁶A depletion. a**, X-chromosomal transcripts are less upregulated upon m⁶A depletion in male mESCs ($P$ value = $1.86 \times 10^{-17}$, two-tailed Wilcoxon rank-sum test). The cumulative fraction of transcripts (RPKM > 1) on individual autosomes (gray) and the X chromosome (orange) that show a given expression fold change (log₂, RNA-seq) upon m⁶A depletion (STM2457, 24 h). Mean expression changes for all autosomes are shown as a black line. Effect sizes (blue) show the shift in medians, expressed as percentage of the average interquartile range (IQR) of autosomal and X-chromosomal genes (see Methods). **b**, X:A expression ratios show a significant reduction upon m⁶A depletion ($P = 1.4 \times 10^{-15}$, two-tailed $t$-test of linear contrasts in mixed effect Gaussian model in log scale). **c**, Differential effects on autosomal and X-chromosomal transcripts already occur after 6 h

of m⁶A depletion. Median fold changes (log₂) of transcripts from autosomes ($n = 19$, gray) and the X chromosome ($n = 1$, orange) estimated by RNA-seq at different time points of m⁶A depletion (STM2457, 3, 6, 9 and 12 h). Boxes represent quartiles, center lines denote medians, and whiskers extend to most extreme values within 1.5 × interquartile range. **d**, Same as **a**, for human primary fibroblasts (STM2457, 9 h). $P$ value = $6.24 \times 10^{-6}$, two-tailed Wilcoxon rank-sum test. Effect sizes are shown as the shift in medians of the two distributions, expressed as percentage of the average IQR of autosomal and X-chromosomal genes (see Methods). **e**, Same as **b**, for human cell lines ($P$ value = 0.0000803 (human fibroblasts), $P$ value = 0.0000379 (HEK293T), $P$ value = 0.0003284 (C643), $P$ value = 0.0002982 (RPE1). $P$ values were calculated as in **a**, with multiple testing correction.

Importantly, this confirmed that all autosomes showed a similar level of m⁶A modifications and that X-chromosomal transcripts were unique in carrying fewer m⁶A sites (Fig. 3d and Extended Data Fig. 6d). As a control, we ensured that this observation was independent of differences in the numbers or lengths of transcripts between chromosomes (see Methods and Extended Data Fig. 6e,f). We observed the same reduction in m⁶A levels on X-chromosomal transcripts in recently published mESC m⁶A-seq2 data[36] (Fig. 3e).

This phenomenon was not restricted to mESCs: we found a similar reduction in m⁶A levels on X-chromosomal transcripts in high-confidence m⁶A sites from mouse heart (female) samples and mouse macrophages (male)[33] (Fig. 3f). The distinct m⁶A patterns also extend to human cells, because human HEK293T (female) and C643 (male) cells displayed a consistent reduction of X-chromosomal m⁶A sites (Fig. 3g). The strength of the reduction was, to some degree, tissue- and species-dependent. Collectively, our results show that X-chromosomal transcripts have fewer m⁶A modifications than do autosomal transcripts across different tissues and cell lines from mice and humans, further supporting that m⁶A-mediated dosage compensation is a conserved mechanism.

### Reduced m⁶A levels are due to GGACH motif depletion
In mammals, m⁶A occurs mainly in a DRACH consensus sequence, with GGACH being the most frequently methylated DRACH motif[23,24]. To test

whether sequence composition has a role in the observed differences in m⁶A levels between chromosomes, we counted the occurrence of GGACH motifs for transcripts on all chromosomes. Remarkably, transcripts on the X chromosome harbored significantly fewer GGACH motifs in their coding sequence (CDS) and 3′ untranslated region (3′ UTR) than did autosomal transcripts (Fig. 4a and Extended Data Fig. 7a). Within 3′ UTRs, autosomal transcripts contained, on average, 3.1 GGACH per kilobase of sequence; this value dropped to 1.7 in X-chromosomal transcripts. This suggests that the lower levels of m⁶A modifications in X-chromosomal transcripts are intrinsically encoded by fewer GGACH motifs. To further investigate this, we compared strongly and weakly methylated DRACH motifs (Extended Data Fig. 7b). Although the strong DRACH motifs were depleted on X-chromosomal transcripts, the weak DRACH motifs were equally abundant on X-chromosomal and autosomal transcripts (Extended Data Fig. 7c,d). This supports the idea that the lower m⁶A levels on X-chromosomal transcripts are a consequence of a reduced number of strongly methylated DRACH motifs. In addition, we observed that, among the GGACH motifs that are present, the fraction that was methylated in mESCs was slightly lower in X-chromosomal than in autosomal transcripts (Fig. 4b and Extended Data Fig. 7e–g), possibly indicating that methylation efficiency of GGACH motifs is also reduced on the X chromosome. To investigate whether this is accompanied by less binding of Mettl3 to X-chromosomal genes, we analyzed published Mettl3 chromatin immunoprecipitation and sequencing (ChIP–seq)

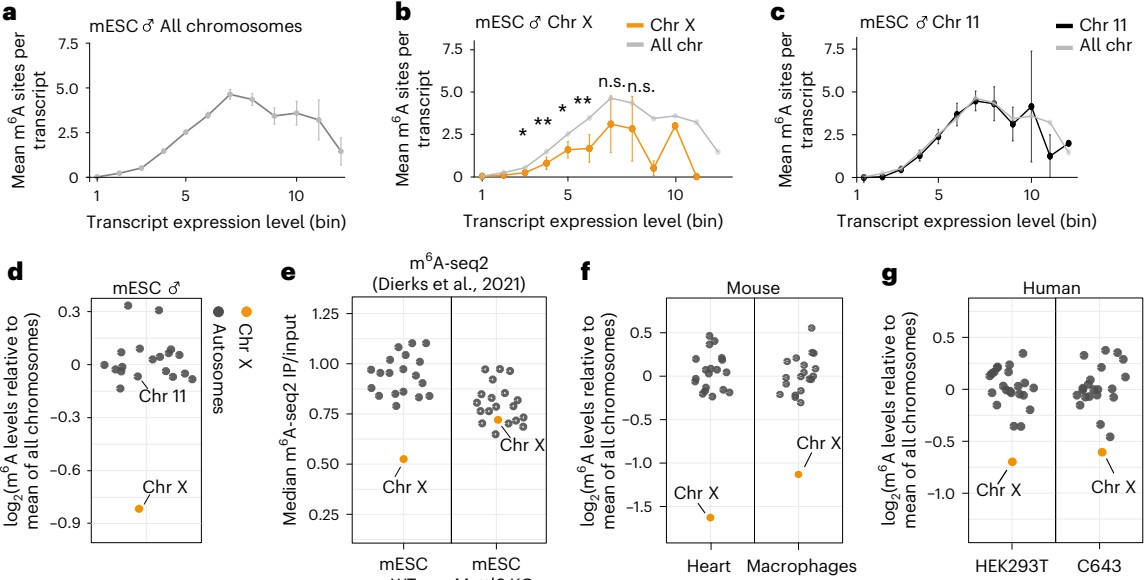

**Fig. 3 | m⁶A sites are reduced on transcripts from the X chromosome. a**, The number of detected m⁶A sites varies with expression level. Mean m⁶A sites per transcript were quantified for transcripts in each expression bin ($n = 12,034$ transcripts, see Extended Data Fig. 6a for $n$ in each bin). Error bars indicate the 95% confidence interval. **b**, X-chromosomal transcripts harbor fewer m⁶A sites across expression levels. Transcripts from the X chromosome (orange, $n = 389$ transcripts) compared with the mean of all chromosomes (gray). The numbers of transcripts in expression bins are shown in Extended Data Figure 6c. Significance values for bins 3–8 are indicated by asterisks (autosomes versus X chromosome, two-tailed Wald tests in a generalized linear model for negative binomial data, multiple testing correction; n.s., not significant; *$P$ value < 0.05, **$P$ value < 0.01). **c**, The m⁶A content of transcripts from chromosome 11 ($n = 1,031$ transcripts) follows the mean of all chromosomes across all expression levels. Transcripts from chromosome 11 (black) compared with the mean of all chromosomes (gray).

Analyses for individual chromosomes are shown in Extended Data Figure 6c. **d–g**, X-chromosomal transcripts have significantly fewer m⁶A sites in male mESCs ($P = 4.1 \times 10^{-9}$, generalized linear model for negative binomial data) (**d**), published m⁶A-seq2 data from mESCs[36] (**e**), mouse heart samples ($P = 8.34 \times 10^{-11}$) and macrophages ($P$ value $= 1.38 \times 10^{-8}$) (**f**), and human HEK293T ($P = 0.000203$) and C643 cell lines ($P$ value $= 0.001030$) (**g**). Mean fold change ($\log_2$) of m⁶A sites per transcript on respective chromosomes relative to all chromosomes (Extended Data Fig. 6d). For mouse data, transcripts of intermediate expression (bins 3–8) are used. For HEK293T data, bins 4–9 were used, and for C643 data, bins 5–10 were used. X-chromosomal and autosomal transcripts are shown in gray and orange, respectively. Chromosomes 11 and X are labeled, for comparison with **b** and **c**. $P$ values for comparisons of autosomal versus X-chromosomal transcripts are as in **b**.

data from mESCs[37]. We observed slightly fewer Mettl3 peaks on the X chromosome, indicating that the co-transcriptional recruitment of Mettl3 to X-chromosomal genes may be reduced (Extended Data Fig. 8a).

Previous reports have suggested that X-to-autosome dosage compensation may be more relevant for certain gene sets than others. For instance, housekeeping genes have been suggested to rely more heavily on upregulation than do tissue-specific genes or recently and independently evolved genes on the X chromosome[5,38,39]. However, we did not observe significant differences in GGACH motifs for different gene sets suggested in the literature (Extended Data Fig. 8b). Furthermore, X-chromosomal genes that have been reported to escape X chromosome inactivation (escaper genes) did not show a significant difference in the number of GGACH motifs compared to other X-chromosomal genes, suggesting that they are equally depleted in m⁶A sites as other X-chromosomal genes[40]. Nonetheless, judging from general variability in GGACH motif content, not all X-chromosomal genes appeared to be equally dependent on dosage compensation. To further investigate this, we performed Gene Ontology (GO) analyses on the 200 genes with the smallest number of GGACH motifs, revealing functionalities related to nucleosomes/DNA packaging and ribosomes being the most significantly enriched (Extended Data Fig. 8c). Indeed, X-chromosomal genes encoding ribosomal proteins and histones harbored almost no GGACH motifs and thereby clearly differed from their autosomal counterparts (Extended Data Fig. 8d), suggesting that proteostasis of these important cellular complexes may be controlled by differential X-to-autosomal m⁶A methylation. This fits with previous reports showing that the majority of the *Minute* phenotypes in *Drosophila* are

caused by haploinsufficiency of ribosomal proteins[41] and that ribosomal protein stoichiometry is tightly controlled in the mouse brain[42].

Next, we wanted to investigate whether GGACH motifs evolved in a sex-chromosome-specific manner. Sex chromosomes are derived from ancestral autosomes. If the selective upregulation of X-chromosomal genes occurs by the reduction of GGACH motifs, outgroup species in which these genes are located on autosomes should not display such a motif disparity. For mammals, the chicken (*Gallus gallus)* is an informative outgroup to investigate the evolution of sex-chromosome expression patterns, because the ancestral eutherian X chromosome corresponds to chromosomes 1 and 4 in chicken[43]. Consequently, the orthologs of X-chromosomal mouse genes are located on autosomes in chicken and are not subjected to sex-chromosome-linked evolutionary changes[17] (Fig. 4c,d). It will be interesting to generate m⁶A maps in different mammalian species to disentangle the contribution of m⁶A to the evolution of mammalian dosage compensation. This will also enable the investigation of X-chromosomal regions of different evolutionary origins, such as X-added region (XAR), X-conserved region (XCR), and pseudoautosomal region (PAR).

To investigate whether the reduction of GGACH motifs on the X chromosome in mouse is a sex-chromosome-linked feature, we compared the GGACH motif content in chicken genes that are orthologous to mouse X-chromosomal or autosomal genes. Of note, given that almost all of these genes reside on autosomes in chicken (Fig. 4d), we observed no difference in GGACH content regardless of whether the orthologs in mouse were on autosomes or the X chromosome (Fig. 4e). This parity of GGACH motifs in the chicken orthologs indicated that

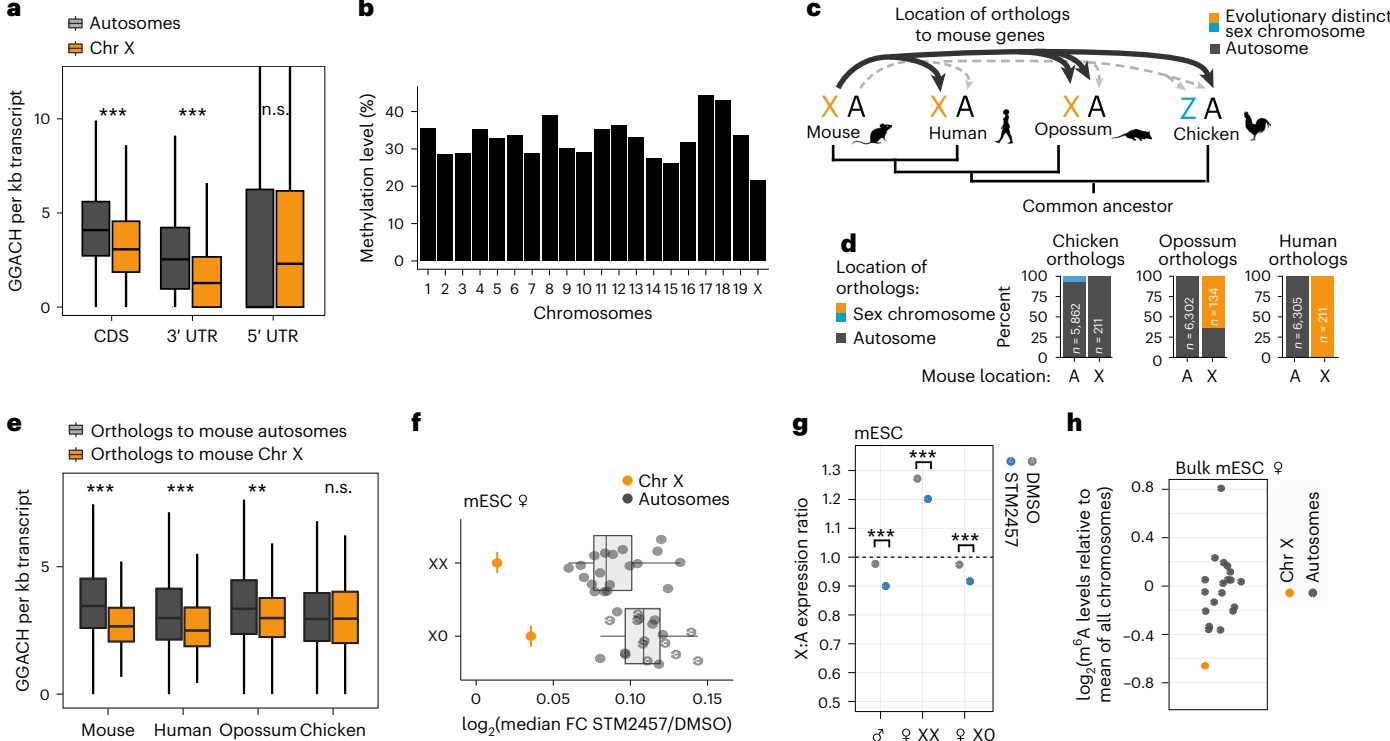

**Fig. 4 | Reduced m⁶A levels on X-chromosomal transcripts are intrinsically encoded. a**, GGACH motifs (normalized to region length) in different transcript regions of autosomal (gray) and X-chromosomal transcripts (orange) in mouse ($P$ value = 1.38 × 10⁻²⁹ (CDS, $n$ = 16,631 annotations), $P$ value = 1.06 × 10⁻⁴⁰ (3′ UTR, $n$ = 16,484 annotations) and 0.2707 (5′ UTR, $n$ = 16,490 annotations), two-tailed Wilcoxon rank-sum test). **b**, Methylation levels of GGACH motifs are slightly reduced on X-chromosomal transcripts. Fraction of m⁶A sites per chromosome with methylation in miCLIP2 data from male mESCs. Boxes represent quartiles, center lines denote medians, and whiskers extend to most extreme values within 1.5 × interquartile range. **c**, Location of mouse X-chromosomal orthologs in human, opossum (*Monodelphis domestica*), and chicken. **d**, Percentage of orthologs of X-chromosomal or autosomal genes in mouse that are located on autosomes or sex chromosomes in human, opossum, and chicken. **e**, GGACH motifs in transcripts (exons) from mouse genes and corresponding orthologs in chicken, opossum, and human ($n$ = 6,520). Orthologs to mouse X-chromosomal

and autosomal genes are indicated in orange and gray, respectively (two-tailed Wilcoxon rank-sum test, *$P$ value < 0.05, **$P$ value < 0.01, ***$P$ value < 0.001, $P$ value = 1.2 × 10⁻¹⁸ (mouse), 2.7 × 10⁻⁶ (human), 0.001227 (opossum), 0.8602 (chicken)). Boxes are as in **a**. **f**, Effects of m⁶A depletion on expression of autosomal and X-chromosomal transcripts in XX and X0 clones of female mESCs ($P$ value = 1.64 × 10⁻¹² and 3.5 × 10⁻¹¹, respectively, two-tailed Wilcoxon rank-sum test, Extended Data Fig. 9a–c). Median fold changes (log₂) of transcripts from autosomes ($n$ = 19, gray) and the X chromosome ($n$ = 1, orange), estimated by RNA-seq after m⁶A depletion (STM2457, 9 h). Boxes are as in **a**. **g**, X:A expression ratios are significantly reduced upon m⁶A depletion ($P$ value = 4.12 × 10⁻¹⁵ (mESC), $P$ value = 2.06 × 10⁻¹¹ (female mESC XX), $P$ value = 1.08 × 10⁻¹⁰ (female mESC X0). $P$ values are as in Figure 2b, multiple testing correction). **h**, Median fold change (log₂) of m⁶A sites per transcript on each chromosome relative to all chromosomes ($P$ = 0.0018, autosomal (gray) versus X-chromosomal (orange) transcripts, two-tailed Wald test in generalized linear mixed model for negative binomial data).

---

the reduced number of GGACH motifs on the mouse X chromosome has evolved specifically as a characteristic of a sex chromosome, in line with the resulting need for X-to-autosome dosage compensation.

## m⁶A contributes to dosage compensation in both sexes

The finding that GGACH motifs are less abundant on the X chromosome suggests that reduced m⁶A levels are an intrinsic feature of X-chromosomal transcripts, which occurs in both sexes independently of X chromosome dosage. To analyze this, we performed RNA-seq experiments in female mESCs in which both X chromosomes are still active and hence dosage compensation is not required. Female mESCs were cultured under standard conditions to ensure that their naive state of pluripotency was maintained[32]. Since female mESCs in cell culture are prone to lose one X chromosome, clonal populations of XX and X0 cells were derived from a given culture plate as matched controls[44–46]. We performed m⁶A depletion (9 h) on 20 colonies and then determined their chromosome content by DNA-seq to choose three XX and three X0 colonies for RNA-seq analyses (Extended Data Fig. 9a–c). Expression analysis revealed that, in female mESCs with two X chromosomes, the median X:A ratio rose above one, indicating that, with two active X chromosomes, genes reach higher levels of expression than

autosomes (Fig. 4g). This supports that one X chromosome is sufficient to obtain a median X:A ratio of 1, whereas two active X chromosomes lead to an excess of X-chromosomal gene expression. Again, the X:A ratio significantly decreased upon m⁶A depletion, further supporting the idea that the depletion of m⁶A impairs X-to-autosome dosage compensation.

We found that, in both XX and X0 colonies, X-chromosomal transcripts significantly differed in their response to m⁶A depletion compared with autosomal transcripts (Fig. 4f and Extended Data Fig. 9d). Subsequently, we identified m⁶A sites in female bulk mESCs using miCLIP2 (ref. 33). In line with our RNA-seq results, and similar to male mESCs, female mESCs had a less m⁶A content on X-chromosomal transcripts (Fig. 4h and Supplementary Table 3). This indicated that, although both X chromosomes are still active in female mESCs, the cells may be able to tolerate higher levels of X-chromosomal transcripts during very early development. The reduced X-chromosomal m⁶A content in female mESCs further supported our finding that the reduced m⁶A levels are intrinsically encoded in the GGACH motif content. Altogether, our results indicate that m⁶A-dependent destabilization of autosomal transcripts also occurs in female mESCs prior to X chromosome inactivation.

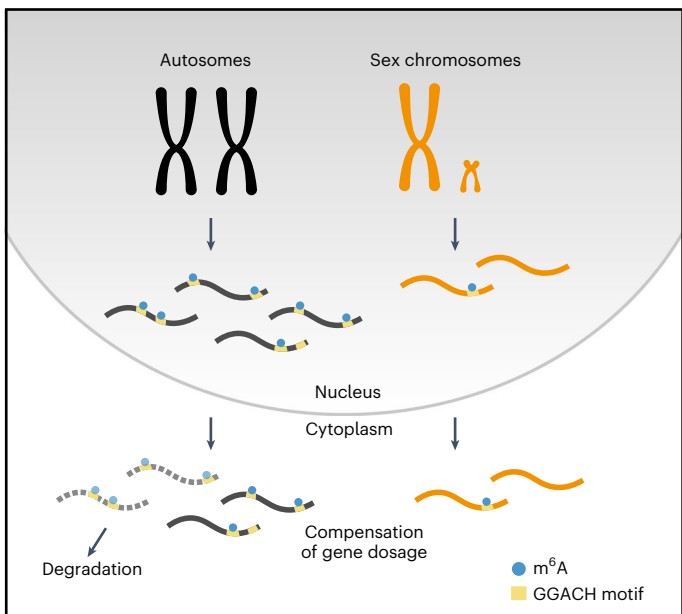

**Fig. 5 | The role of m⁶A in X-to-autosome dosage compensation.** m⁶A acts as a selective degradation signal on autosomal transcripts and thereby contributes to X-to-autosome dosage compensation. Transcripts from the autosomes are transcribed from two active chromosomes, leading to higher transcript copy numbers per autosomal gene than for X-chromosomal genes. m⁶A is selectively enriched on transcripts from autosomes, leading to their destabilization and degradation. Because m⁶A is not enriched on X-chromosomal transcripts, this leads to an equal dosage between autosomal and X-chromosomal transcripts. m⁶A thereby contributes to X-to-autosome dosage compensation.

## Discussion

X-chromosomal genes are expressed from only one active chromosome copy in mice and humans. To balance the genetic input between dual-copy autosomal and single-copy X-chromosomal transcripts, Susumo Ohno hypothesized more than 50 years ago that compensating mechanisms are required for balancing gene expression[3]. Here, we uncover that differential m⁶A methylation adds a layer of complexity to X-to-autosomal dosage compensation in eutherian mammals. This causes a global destabilization of m⁶A-containing autosomal transcripts, while X-chromosomal transcripts bypass this regulatory mechanism (Fig. 5). Importantly, we show that the inhibition of m⁶A methylation predominantly stabilizes autosomal transcripts and thereby affects the X-to-autosome balance of gene expression.

Several sex-chromosome-compensating mechanisms identified so far, including X inactivation in mammals, XX dampening in *Caenorhabditis elegans*, and X-chromosomal upregulation in *Drosophila melanogaster*, act on the chromatin environment of the sex chromosomes and have been shown to influence RNA polymerase II occupancy and transcription of X-chromosomal genes[7,16,47–52]. In addition, RNA-regulatory mechanisms, including higher RNA stability and translational efficiency of X-chromosomal transcripts, as well as an enrichment of NMD targets and microRNA-targeting sites among autosomal transcripts, have been described as X-to-autosome dosage-compensation pathways[4,6,8–10,53,54].

In contrast to the previously described regulatory mechanisms, m⁶A-mediated dosage compensation acts globally at the epitranscriptomic level and adds an additional layer of regulation to X-to-autosome dosage compensation. Importantly, by inhibiting m⁶A methylation, we can interfere experimentally with this process, thereby partly disrupting X-to-autosomal dosage compensation. We propose that m⁶A-mediated dosage compensation is co-transcriptionally initiated

in the nucleus, where m⁶A deposition is catalyzed[22], and then executed in the cytoplasm, where m⁶A-modified transcripts are presumably degraded[25–27]. Several reasons as to why mammals evolved an epitranscriptomic mechanism for dosage compensation are conceivable. For instance, such a mechanism might be most compatible with the epigenetically installed X-chromosome inactivation in females. By contrast, installing two epigenetic pathways that antagonistically affect the two X chromosomes at the same time might be more difficult to evolve. Interestingly, X chromosome inactivation has also been shown to depend on m⁶A methylation of the non-coding RNA XIST[55], suggesting that dosage compensation and X chromosome inactivation might be coordinated. Furthermore, RNA-based gene regulation is often used to fine-tune gene expression[56]. This meets the needs of dosage compensation, which requires a maximum of a twofold expression regulation. Hence, m⁶A regulation might be ideally suited to establish and maintain small changes. Finally, RNA-based mechanisms offer an elegant means to uncouple X-to-autosome dosage compensation from other levels of gene expression regulation. Because RNA-based mechanisms globally affect all X-chromosomal and autosomal transcripts that are expressed at a given moment, it facilitates genetic equilibrium between chromosomes without interfering with transcriptional regulation per se; for instance, cell-type-specific regulation would remain unaffected.

Our data suggest that differential m⁶A methylation evolved through a loss and/or gain of m⁶A consensus motifs (GGACH) on X-chromosomal and autosomal transcripts during mammalian sex-chromosome evolution, respectively. This means that m⁶A dosage compensation is hardcoded in the individual transcripts and consistently acts on both male and female cells. On top of this, there could be mechanisms that globally modulate m⁶A methylation on X-chromosomal or autosomal transcripts, such as Mettl3 recruitment through the chromatin mark trimethylated histone H3 K36 (ref. [57]) or a local sequestration of Mettl3 through LINE-1 transposons that are heavily m⁶A-methylated and enriched on the X chromosome[58,59]. Moreover, the m⁶A-mediated effects may be linked to the previously suggested role of NMD in X-to-autosome dosage compensation[9], since the NMD key factor UPF1 has been found to be associated with YTHDF2 (ref. [60]).

An exciting question for future research is how the hardcoding of m⁶A-mediated dosage compensation evolved. Here, the short and redundant m⁶A consensus sequence could enable easy generation or removal of consensus sequences. However, why would evolution globally select for m⁶A sites to differentially affect transcripts from different chromosomes? We think that using predominantly hardcoded m⁶A sites allows global modulation of dosage compensation, for instance through the overall methylation levels or the expression of the m⁶A reader proteins that control RNA decay under certain conditions. Although m⁶A levels appear to be relatively stable between tissues in mice and humans[61], it will be interesting to decipher how dosage compensation is globally modulated in different tissues, developmental stages, and pathological conditions.

## Online content

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

## Methods

### Cell culture

All cell culture was performed in a humidified incubator at 37 °C and 5% $CO_2$. All cell lines were routinely monitored for mycoplasma contamination.

Parental male and female mESCs[32,44] were provided by D. Dominissini (Tel Aviv University, Israel) and E. Heard (EMBL Heidelberg, Germany). mESC lines were further authenticated by RNA-seq. Standard tissue culture was performed in 2i/LIF medium. Briefly, 235 ml of each DMEM/F12 and neurobasal (Gibco, 21331020, 21103049) was mixed with 7.5 ml BSA solution (7.5%, Thermo Fisher Scientific, 11500496), 5 ml penicillin–streptomycin (P/S, Thermo Fisher Scientific, 10378016), 2 mM L-glutamine (Thermo Fisher Scientific, 25030024), 100 µM β-mercaptoethanol (Gibco, 21985023), 5 ml mM nonessential amino acids (Gibco, 11140050), 2.5 ml N-2 supplement (Gibco, 17502048), 5 ml B-27 supplement (Gibco, 17504044), 3 µM CHIR99021 (Sigma, SML1046), 1 µM PD 0325901 (Biomol, 13034-1), 10 ng ml$^{-1}$ LIF (IMB Protein Production Core Facility). Cell culture dishes were coated using 0.1% gelatine (Sigma, ES-006-B). The medium was exchanged every day, and cells were passaged every second day. Single colonies of female mESCs were picked under the microscope using a pipette tip and cultured under standard conditions in 96-well plates until confluency was reached.

HEK293T (ATCC, CRL-3216) and C643 (CLS, RRID: CVCL_5969) cells were cultured in DMEM (Thermo Fisher Scientific, 21969035) supplemented with 10% fetal bovine serum (FBS, Pan Biotech, P40-47500), 1% P/S (Thermo Fisher Scientific, 10378016), and 1% L-glutamine. RPE1 (ATCC, CRL-4000) cells were cultured in DMEM/F12 (Thermo Fisher Scientific, 21331020) supplemented with 10% FBS (Pan Biotech, P40-47500), 1% P/S (Thermo Fisher Scientific, 10378016), 1% L-glutamine, and 0.04% hygromycin B (Thermo Fisher Scientific, 10453982).

Human primary dermal fibroblasts were provided by S. Schweiger (University Medicine Mainz, Germany). Cells were grown in IMDM medium (Thermo Fisher Scientific, 12440053) supplemented with 15% FBS and 1% P/S.

### Primary human dermal fibroblasts derivation

Primary human dermal fibroblasts were isolated from skin punch biopsies obtained at the Children's Hospital of the University Medical Center in Mainz, Germany, as previously described with small adjustments[62]. Briefly, 4-mm skin biopsies were processed in small pieces and transferred into a 6-well plate coated with 0.1% gelatine. DMEM (Thermo Fisher Scientific, 21969035) supplemented with 20% FBS (Pan Biotech, P40-47500) and 1% P/S (Thermo Fisher Scientific, 10378016) was used for culturing the skin biopsies, and medium was exchanged every other day. After 3–4 weeks, when the 6-well plate was full of dermal fibroblasts that had migrated out of the skin biopsies, cells were transferred to T75 flasks and cultured in standard conditions. Human dermal fibroblasts were further expanded or frozen in liquid nitrogen for long-term storage. Ethical approval by the local ethical committee was obtained (no. 4485), and consent for research use was obtained in an anonymized way.

### Mettl3 inhibitor treatment

For acute m$^6$A depletion in mESCs, the Mettl3 inhibitor STM2457 (STORM Therapeutics) was used. Cells were treated with medium supplemented with 20 µM STM2457 in DMSO 0.2% (vol/vol) or with DMSO 0.2% (vol/vol) alone as control. m$^6$A depletion was monitored by liquid chromatography with tandem mass spectrometry (LC–MS/MS). After 3–24 h of treatment, cells were washed twice with ice-cold 1× PBS and collected on ice for further analysis

### RNA isolation and poly(A) selection

Cells were washed twice with ice-cold 1× PBS and collected on ice. For total RNA isolation, the RNeasy Plus Mini Kit (Qiagen, 74136) was used,

following the manufacturer's instructions. For poly(A) selection, Oligo d(T)25 Magnetic Beads (Thermo Fisher Scientific, 61002) were used, following the manufacturer's instructions.

### qPCR

For quantification of mRNA levels, 500 ng total RNA was reverse transcribed into complementary DNA (cDNA) using the RevertAid Reverse Transcriptase (Thermo Fisher Scientific, 10161310) using oligo(dT)18 primer (Thermo Fisher Scientific, SO131), following the manufacturer's instructions. In accordance with the manufacturer's instructions, qPCR reactions were performed in technical triplicates using the Luminaris HiGreen qPCR Master Mix, low ROX (Thermo Fisher Scientific, K0971), with forward and reverse primer (0.3 µM each) and 2 µl of 1:10 diluted cDNA as template. All qPCR reactions were run on a ViiA 7 Real-Time PCR System (Applied Biosystems). All qPCR primers are listed in Supplementary Table 4.

### LC–MS/MS

LC/MS-MS experiments were performed as described in ref. 33. For all samples, quantification involved biological duplicates and averaged values of m$^6$A normalized to A, and the respective s.d. values are shown.

### SLAM-seq

**Cell viability for optimization.** To determine the 10% maximal inhibitory concentration in a determined time window ($IC_{10,ti}$), the Cell Viability Titration Module from LeXogen (059.24) was used, following the manufacturer's recommended protocol. In brief, 5,000 cells were plated in a 96-well plate 1 d prior to the experiment. Cells were incubated for 24 h in media supplemented with varying s$^4$U concentrations. For optimal incorporation, the s$^4$U-supplemented media were exchanged every 3 h. Cell viability was assessed using the CellTiter-Glo Luminescent Cell Viability Assay Kit from Promega (G7570), following the manufacturer's recommended protocol. The luminescence was measured using Tecan Infinite M200 Pro plate reader. The cell doubling time of male mESCs in the presence of 100 µM s$^4$U was 13.3 h, as determined by cell counting.

**SLAM-seq experiment.** mRNA half-lives were determined by SLAM-seq using the Catabolic Kinetics LeXogen Kit (062.24). In brief, mESCs were seeded 1 d before the experiment in a 24-well plate to reach full confluency, according to the doubling time, at the time of sample collection. The metabolic labeling was performed by addition of 100 µM s$^4$U to the mESC medium for 24 h. The medium was exchanged every 3 h. After the metabolic labeling, cells were washed twice with 1× PBS, and fresh medium was supplemented with a 100× excess of uridine. At time points increasing at a 1.5× rate, medium was removed and cells were directly lysed in TRIzol (Thermo Fisher Scientific,15596026) reagent in reducing conditions. Total RNA was resuspended in the elution buffer in the Lexogen catabolic kit. The iodoacetamide treatment was performed using 5 µg of RNA. The library preparation for sequencing was performed using the QuantSeq 3′ mRNA-Seq Library Prep Kit for Illumina (FWD) from Lexogen, following the recommended protocol.

For stable m$^6$A depletion, STM2457 or DMSO was supplemented 6 h prior to the uridine chase. The media for the uridine chase were supplemented with STM2457 and DMSO for continuous m$^6$A depletion.

**SLAM-seq library preparation.** Library preparation for next-generation sequencing was performed with QuantSeq 3′ mRNA-Seq Library Prep Kit FWD (Lexogen, 015), following the manufacturer's standard protocol (015UG009V0252). Prepared libraries were profiled on a 2100 Bioanalyzer (Agilent Technologies) and quantified using the Qubit dsDNA HS Assay Kit, in a Qubit 2.0 Fluorometer (Life Technologies). All samples were pooled together in an equimolar ratio and sequenced on an Illumina NextSeq 500 sequencing device using three High Output flow cells as 84-nt single-end reads.

**Data processing.** Published SLAM-seq data were taken from ref. 28. 3′ UTR annotations were taken from ref. 28 and filtered to match the GENCODE annotation[63] release M23. Non-overlapping annotations were discarded.

Raw data were quality checked using FastQC (v0.11.8) (https://www.bioinformatics.babraham.ac.uk/projects/fastqc/). Sequencing data were processed using SLAM-DUNK (v0.4.3)[64] with the following parameters: mapping was performed allowing multiple mapping to up to 100 genomic positions for a given read (-n 100). Reads were filtered using SLAM-DUNK -filter with default parameters. For annotation of single-nucleotide polymorphisms (SNPs), all unlabeled samples were merged and SNPs were called using SLAM-DUNK snp with default parameters and -f 0.2. Transition rates were calculated using SLAM-DUNK count with default parameters, providing the SNP annotation of unlabeled samples (-v). If more than one 3′ UTR per gene remained, they were collapsed using SLAM-DUNK collapse[64]. Only genes on canonical chromosomes 1–19 and X were considered.

**Principal component analysis.** Principal component analysis (PCA) of SLAM-seq data was performed by estimating size factors on the basis of read counts using the R/Bioconductor package DESeq2 (ref. 65) (v1.26.0) in an R environment (v3.6.0). PCA was based on the number of T-to-C reads per gene for 500 genes with the highest variance, corrected by the estimated size factors.

**Incorporation rate.** $s^4U$ incorporation rates were calculated by dividing the number of T-to-C conversions on Ts for each 3′ UTR by the overall T coverage.

**Half-life calculation.** To calculate mRNA half-lives, T-to-C background conversion rates (no $s^4U$ labeling) were subtracted from T-to-C conversion rates of $s^4U$-labeled data. Only 3′ UTRs with reads covering over 100 Ts (T-coverage > 100) were kept (Extended Data Fig. 2d). For each time point, T-to-C conversion rates were normalized to the time point after 24 h of $s^4U$ labelling (that is, the onset of the uridine chase), which corresponds to the highest amount of $s^4U$ incorporation in the RNA (24 h $s^4U$ labelling, T0) and fitted using an exponential decay model for a first-order reaction using the lm.package (as described in ref. 28, adapted from ref. 66). Half-lives of >18 h (1.5 times of the last time point) and <0.67 h, as well as fitted values with a residual s.e. of >0.3, were filtered out (Extended Data Fig. 2e). Only transcripts with a valid half-life calculation in both conditions were kept for further analysis. For statistical analysis of half-life fold changes, see Supplementary Methods.

**RNA-seq library preparation and data processing**

**RNA-seq library preparation.** RNA-seq library preparation was performed with Illumina's Stranded mRNA Prep Ligation Kit following the Stranded mRNA Prep Ligation Reference Guide (June 2020) (document no. 1000000124518 v00). Libraries were profiled on a 2100 Bioanalyzer (Agilent Technologies) and quantified using the Qubit dsDNA HS Assay Kit (Thermo Fisher Scientific, Q32851) in a Qubit 2.0 Fluorometer (Life Technologies), following the manufacturer's recommended protocols. Samples were pooled in equimolar ratios and sequenced on an Illumina NextSeq 500 sequencing device with one or two dark cycles upfront as 79-, 80- or 155-nt single-end reads.

**Data processing.** Basic quality controls were done for all RNA-seq samples using FastQC (v0.11.8) (https://www.bioinformatics.babraham.ac.uk/projects/fastqc/). Prior to mapping, possible remaining adapter sequences were trimmed using Cutadapt[67] (v1.18). A minimal overlap of 3 nt between read and adapter was required, and only reads with a length of at least 50 nt after trimming (--minimum-length 50) were kept for further analysis. For samples sequenced with only one

dark cycle at the start of the reads, an additional 1 nt was trimmed at their 5′ ends (--cut 1).

Reads were mapped using STAR[68] (v2.7.3a), allowing up to 4% of the mapped bases to be mismatched (--outFilterMismatchNoverLmax 0.04 --outFilterMismatchNmax 999) and a splice junction overhang (--sjdbOverhang) of 1 nt less than the maximal read length. Genome assembly and annotation of GENCODE[63] release 31 (human) or release M23 (mouse) were used during mapping. In the case that ERCC spike-ins were added during library preparation, their sequences and annotation (http://tools.thermofisher.com/content/sfs/manuals/ERCC92.zip) were used in combination with those from GENCODE. Subsequently, secondary hits were removed using SAMtools[69] (v1.9). Exonic reads per gene were counted using featureCounts from the Subread tool suite[70] (v2.0.0) with non-default parameter --donotsort -s2.

**Differential gene expression analysis.** Differential gene expression between conditions was performed using the R/Bioconductor package DESeq2 (v1.34.0)[65] in the R environment (v4.1.2; https://www.R-project.org/). DESeq2 was used with a significance threshold of adjusted $P$ value < 0.01 (which was also used to optimize independent filtering). Because normalization to total transcript abundance can introduce biases, especially when the majority of genes are affected by the treatment, we included spike-ins in our initial RNA-seq dataset. As an alternative normalization strategy to spike-ins, we tested 100 randomly chosen genes without any $m^6A$ sites but noticeable expression (reads per kilobase per million fragments mapped (RPKM) > 10) for normalization. To validate this normalization approach, the calculated fold changes were compared with spike-in-normalized data. Because the correlation between both normalization strategies was very high, we used the 100 genes for normalization in all further analyses (Extended Data Fig. 3b). For RNA-seq expression change analysis, see Supplementary Methods and Supplementary Table 5.

**miCLIP2**
miCLIP2 experiments were performed as described in ref. 33. For a detailed description of analyses, see Supplementary Methods.

**Quantification of $m^6A$ sites in transcripts**
$m^6A$ sites from miCLIP2 for male mESCs, mouse heart samples, mouse macrophages, and human HEK293T and C643 cells were taken from ref. 33 (Gene Expression Omnibus (GEO) accession number GSE163500). $m^6A$ sites were predicted using $m^6A$boost as described in ref. 33. For miCLIP2 mouse heart data, only $m^6A$ sites that were predicted by $m^6A$boost in both datasets (1 μg and 300 ng) were considered for the analysis.

**Comparison of $m^6A$ sites per transcripts.** Numbers of $m^6A$ sites were counted for each protein-coding transcript. Only transcripts on canonical chromosomes 1–19 and X were considered. To account for expression differences, transcripts were stratified according to their expression levels on the basis of the respective miCLIP2 data. Expression levels were estimated using htseq-count[71] (v0.11.1) and genome annotation of GENCODE[63] release M23 on the truncation reads from miCLIP2 data (noC2T reads)[33]. The derived transcript per million (TPM) values for all replicates ($n = 3$) were averaged, $\log_{10}$-transformed, and used to stratify all transcripts into 12 equal-width bins (step size of $\log_{10}(\text{TPM}) = 0.25$), collecting all transcripts with $\log_{10}(\text{TPM}) < 0.5$ or > 3 into the outer bins (Extended Data Fig. 6a). A minimum of TPM > 1 was set. For each expression bin, the mean and 95% confidence interval of the number of $m^6A$ sites per transcript were calculated (Fig. 3a–c and Extended Data Fig. 6c). To estimate the fold change of $m^6A$ sites per chromosome compared with all other chromosomes (Fig. 3d,f,g), only transcripts with intermediate expression (bins 3–8) were taken into account (mouse). For HEK293T data, bins 4–9 were used, and for C643 data, bins 5–10 were used. For each bin, the difference of $m^6A$ levels of a chromosome relative to all chromosomes was calculated. For this,

the mean number of m⁶A sites on transcripts of the chromosome was divided by the mean number of m⁶A sites on transcripts of all chromosomes in the given bin (for example, orange dots (X chromosome) over gray dots (all transcripts) in Figure 3b). This resulted in a fold change of m⁶A sites of each chromosome over all chromosomes for each of the six considered bins (Extended Data Fig. 6d). For comparison with other chromosomes (Fig. 3d,f,g), the mean fold change per chromosome over all expression bins was calculated (Extended Data Fig. 6d, red dot).

**Control for transcript-length biases.** To exclude biases from different transcript lengths, we repeated the analysis using only m⁶A sites within a 201-nt window (−50 nt to +150 nt) around the stop codon, in which a large fraction of m⁶A sites accumulate[23]. To obtain stop codon positions, transcript annotations from GENCODE[63] release M23 were filtered for the following parameters: transcript support level ≤ 3, level ≤ 2, and the presence of a Consensus Coding Sequence (CCDS) ID (ccdsid). If more than one transcript per gene remained, the longer isoform was chosen. Repeating the analyses with this subset, as described above, supported our observation that X-chromosomal transcripts harbor fewer m⁶A sites without being influenced by differences in transcript lengths (Extended Data Fig. 6e).

**Subsampling of transcripts in expression bins.** To account for potential biases from different numbers of transcripts in the expression bins for each chromosome, we randomly picked 30 genes for each expression bin (using bins 3–5, 90 genes in total) and calculated the fold change of m⁶A content on transcripts for each chromosome compared with all other chromosomes, as described above. The procedure was repeated 100 times. The distribution of resulting fold change values supports that X-chromosomal transcripts harbor fewer m⁶A sites, regardless of the number of transcripts considered (Extended Data Fig. 6f).

**Statistical analysis of m⁶A sites in transcripts.** See Supplementary Methods and Supplementary Table 6.

### Analysis of published m⁶A-seq2 data
Published m⁶A-seq2 data for wild-type and *Mettl3* KO mESCs were retrieved from ref. 36. We used the 'gene index,' that is, the ratio of m⁶A IP values over IP for whole genes, as a measure of the transcripts methylation level, as described in ref. 36 (Fig. 3e). Chromosome locations of the genes (*n* = 6,278) were assigned using the provided gene name in the R/Bioconductor package biomaRt in the R environment[72,73].

### DRACH motif analyses
**GGACH motifs in mouse transcripts.** Mouse transcript annotations from GENCODE[63] release M23 were filtered for the following parameters: transcript support level ≤ 3, level ≤ 2, and the presence of a CCDS ID. If more than one transcript annotation remained for a gene, the longest transcript was chosen. Different transcript regions (3′ UTR, 5′ UTR, CDS) were grouped per gene, and GGACH motifs were counted per base pair in different transcript regions, for example, the sum of GGACH motifs in CDS fragments of a given gene was divided by sum of CDS fragment lengths.

**GGACH motifs in chicken, opossum, and human orthologs.** Orthologs of mouse genes in chicken (*Gallus gallus*), human (*Homo sapiens*), and opossum (*Monodelphis domestica*) were retrieved from the orthologous matrix (OMA) browser[74] (accessed on 21 March 2022, for opossum 28 July 22). Only one-to-one orthologs were kept. Genes were filtered to have orthologs in all three species (*n* = 6,520). Then, numbers of GGACH motifs per base pair of all protein-coding exons were quantified on the basis of GENCODE annotation (release 31)[63] for human and ENSEMBL annotation (release 107, genome assembly GRCg6a)[75] for chicken and opossum annotation (ASM229v1). GGACH motifs per base pair were quantified and visualized as described above.

**Estimation of methylation levels.** See Supplementary Methods.

**GGACH in gene sets from literature.** Independently evolved gene sets and genes with or without an ortholog on the human X chromosome were taken from ref. 39. Escaper genes were taken from ref. 16. Testis-specific genes were taken from ref. 5. Genes from the X-added XAR and XCR were annotated by identifying X-chromosomal genes in mouse with the location of chicken orthologs on chromosome 1 (XAR) and chromosome 4 (XCR).

### ChIP–seq analysis
ChIP–seq peaks were obtained from ref. 37. The numbers of peaks per chromosome were divided by chromosome lengths. To calculate the peak ratio per chromosome compared with all other chromosomes, the normalized peak number per chromosome was divided by the median peak number of all chromosomes.

### GO analysis
GO term enrichment was performed using the enrichGO function of clusterProfiler[76] (v.4.2.2). Cellular components (ont = 'CC') were enriched using a *P* value cut-off of 0.01 and a *q* value cut-off of 0.05, and *P* values were corrected using Benjamini–Hochberg correction (pAdjustMethod = 'BH').

### DNA-seq to determine copy number variation
See Supplementary Methods.

### Statistics and reproducibility
All statistical analyses were performed using R. All boxplots in this study are defined as follows: boxes represent quartiles, center lines denote medians, and whiskers extend to most extreme values within 1.5 × interquartile range. All statistical tests performed in this study were two-tailed. All indicated replicate numbers refer to independent biological replicates. No statistical method was used to predetermine sample size. The experiments were not randomized. No data were excluded from the analysis, unless stated otherwise. The investigators were not blinded during allocation in experiments or to outcome assessment.

### Reporting summary
Further information on research design is available in the Nature Portfolio Reporting Summary linked to this article.

## Data availability
All high-throughput sequencing datasets generated in this study were submitted to the Gene Expression Omnibus (GEO) under the SuperSeries accession GSE203653. RNA-seq data for human primary fibroblasts are available via the EGA European Genome-Phenome Archive under the accession number EGAS00001007112. Source data are provided with this paper.

## Code availability
The scripts used to process the files are accessible under the GitHub repository located at: github.com/crueckle/Rueckle_et_al_2023.

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

## Acknowledgements

We thank E. Heard (EMBL Heidelberg, Germany) for providing female mESCs (TX1072). We thank D. Dominissini (Tel Aviv University, Israel) for providing male mESCs. We gratefully acknowledge the support of the IMB Genomics Core Facility and the use of the NextSeq 500 (funded by the Deutsche Forschungsgemeinschaft (DFG, German Research Foundation) – INST 247/870-1FUGG) and members of the Genomics and Bioinformatics Core Facilities for technical support. C.R., N.K., K.T., M.B., and M.P. were supported by the International PhD Programme on Gene Regulation, Epigenetics & Genome Stability, Mainz, Germany. Animal shapes in Figure 4c were obtained from PhyloPic and are used under the Creative Common Attribution-NonCommercial-ShareAlike 3.0 Unported license. This work was supported by the Deutsche Forschungsgemeinschaft (DFG, German Research Foundation) (SPP 1935 (Projektnummer 273941853), KO4566/3-2 and TRR 319 (Projektnummer 439669440) to J.K.; SPP 1935 (Projektnummer 273941853), ZA881/5-2 to K.Z.; INST 247/870-1FUGG). The funders had no role in study design, data collection and analysis, decision to publish, or preparation of the manuscript.

## Author contributions

The majority of bioinformatic analyses were performed by C.R. with the help of A.B. and Y.Z. The majority of experiments were performed by N.K. with the help of M.F.B., P.H.-K., K.T., M.B., M.P., and M.M. Statistical analyses were performed by C.R. and F.K. C.R., N.K., M.F.B., K.Z., C.I.K.V., and J.K. designed the study and wrote the manuscript. C.R., N.K., M.F.B., A.B., Y.Z., P.H.-K., K.T., F.K., M.B., M.P., M.M., S.S., C.N., O.R., K.Z., C.I.K.V., and J.K. contributed to the design of the study and read and commented on the manuscript.

## Competing interests

O.R. is an employee of STORM Therapeutics Ltd. The other authors declare that they have no competing interests.

## Additional information

**Extended data** is available for this paper at https://doi.org/10.1038/s41594-023-00997-7.

**Correspondence and requests for materials** should be addressed to Julian König.

**Peer review information** *Nature Structural & Molecular Biology* thanks the anonymous reviewers for their contribution to the peer review of this work. Sara Osman and Dimitris Typas were the primary editors on this article and managed its editorial process and peer review in collaboration with the rest of the editorial team. Peer reviewer reports are available.

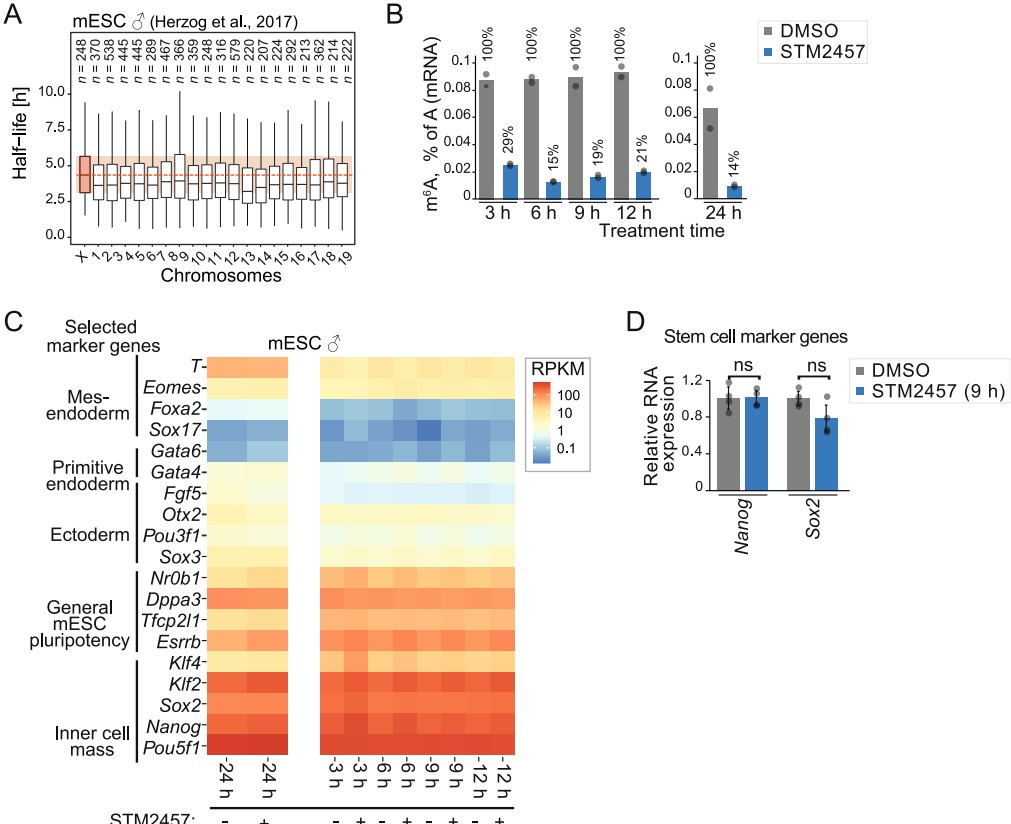

**Extended Data Fig. 1 | Mettl3 inhibitor treatment of mouse embryonic stem cells (mESC) depletes m⁶A levels. a**. X-chromosomal transcripts are more stable than autosomal transcripts (median half-life = 3.72 h [autosomes] vs. 4.35 h [X chromosome], *P* value = 1.02e-05, two-sided Wilcoxon rank-sum test). Distribution of half-lives from published SLAM-seq data for mESC for transcripts on each individual chromosome. Dashed red line and red box indicate median and inter-quartile range of X-chromosomal transcripts, respectively, for comparison. Boxes represent quartiles, centre lines denote medians, and whiskers extend to most extreme values within 1.5× interquartile range. **b**. Time course experiments shows that treatment of male mESC with the Mettl3 inhibitor (STM2457, 20 μM) results in a gradual reduction of m⁶A levels on mRNAs. m⁶A levels were measured by liquid chromatography-tandem mass spectrometry (LC-MS/MS) for poly(A) + RNA from m⁶A-depleted (STM2457, 3–24 h) and

control conditions. Quantification of m⁶A as percent of A in poly(A) + RNA. *n* = 2 independent biological replicates. **c**. Expression levels of marker genes confirm the pluripotent state of the male mESC throughout the time course experiment. Gene expression levels (RNA-seq) are shown as reads per kilobase of transcript per million mapped reads (RPKM, mean over all replicates, log₁₀) in m⁶A-depleted (STM2457, 3–24 h) and control conditions. **d**. Quantitative real-time PCR (qPCR) to quantify expression changes of stem cell marker genes in m⁶A-depleted (STM2457, 9 h) and control conditions. Normalised C_T values (ΔC_T, normalised to *Gapdh* expression) are compared between conditions. Fold changes are displayed as mean s.d.m., two-sided Student's *t*-test on log₂-transformed data, *n* = 4 independent biological samples, ns, not significant. *P* value = 0.8 [*Sox2*]; 0.96 [*Nanog*].

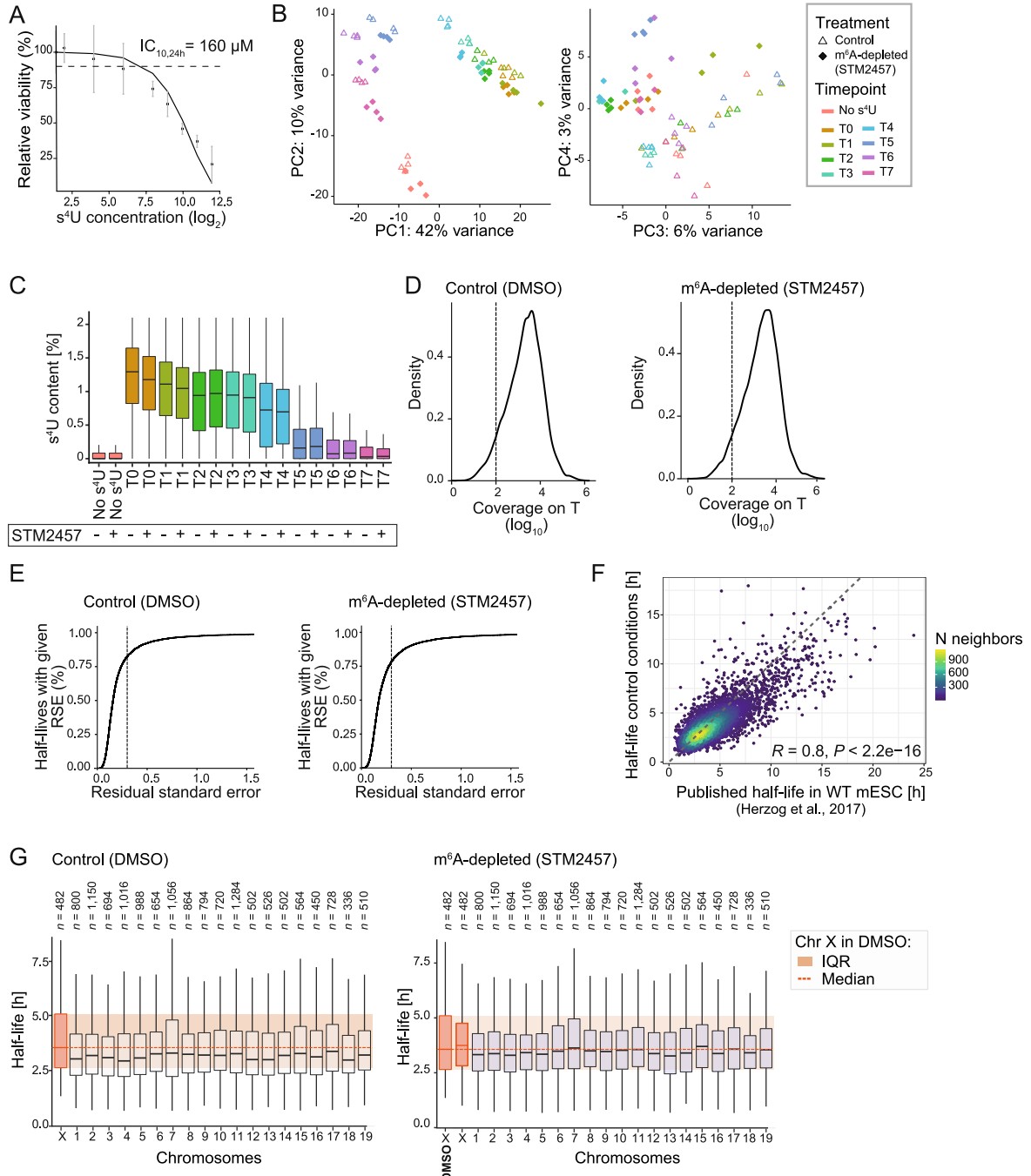

**Extended Data Fig. 2 | SLAM-seq measures mRNA half-lives in mESC.**
**a**. Cell viability assessed for male mESC cultured with s4U for 2 ± 4 h in varying concentrations (x-axis, log2-transformed). Viability of labelled cells in relation to unlabelled cells is shown as mean ± s.d.m., $n = 3$ biologically independent samples. IC10,24h is indicated as dashed line. **b**. Principal component analysis of SLAM-seq replicates based on numbers of reads with T-to-C conversions. Principal component (PC) 1 and PC2 (left) separate the different timepoints of the experiment (colours), PC3 and PC4 (right), separate control and m6A-depleted conditions (symbols). **c**. T-to-C conversions on T's by the overall T coverage per 3' UTR. Maximum s4U rate is achieved after 24 h of labelling (T0) and steadily decreases after s4U washout and uridine chase (T1-T7). Unlabelled samples (No s4U) are shown for comparison. $n = 21,527$ UTRs with incorporation rates per replicate. Boxes represent quartiles, centre lines denote medians, and whiskers extend to most extreme values within 1.5× interquartile range. **d**. Expression estimates based on log10-transformed coverage on T's per 3' UTR (mean over all replicates and timepoints per condition). Only 3' UTRs with SLAM-seq reads

covering at least 100 T's (indicated by dotted line) were used for subsequent fitting. **e**. Cumulative distribution of the goodness-of-fit (residual standard error, RSE) of half-lives calculated from SLAM-seq data. Dotted lines indicate filtering cut-off (RSE > 0.3). **f**. Correlation of half-lives determined in this study (male mESC, control condition) with previously published half-lives in male mESC (two-sided Pearson correlation coefficient [$R$] = 0.8, $P$ value < 2.2e-16). **g**. Distribution of half-lives of transcripts on individual chromosomes in control (left) or m6A-depleted conditions (right). In control conditions, half-lives of X-chromosomal transcripts differ significantly from autosomal transcripts (median half-life 3.19 h [autosomes] vs. 3.57 [X chromosome], $P$ value = 7.63e-05, two-sided Wilcoxon rank-sum test). In m6A-depleted conditions, autosomal transcript half-lives approximate X-chromosomal transcript half-lives in control conditions ($P$ value = 0.06228, two-sided Wilcoxon rank-sum test). Red lines and boxes indicate median and interquartile range, respectively, of half-lives of X-chromosomal transcripts in control conditions. Boxes as in **c**.

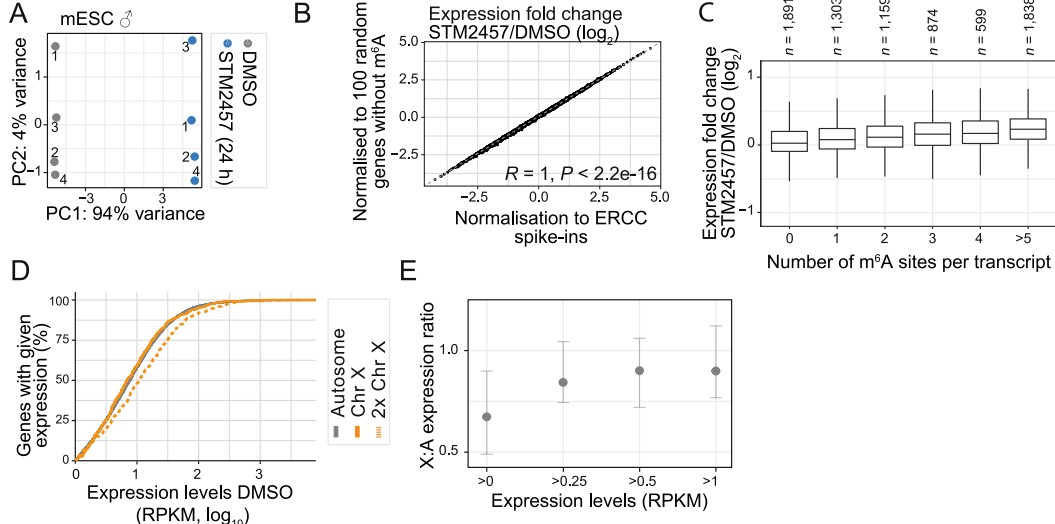

**Extended Data Fig. 3 | RNA-seq upon m⁶A depletion reveals upregulation of autosomal but not X-chromosomal transcripts. a**. Principal component analysis indicates high reproducibility of RNA-seq data for male mESC in control and m⁶A-depleted conditions (STM2457, 24 h, 4 replicates per condition, total of 398 million uniquely mapped reads). Replicate number given next to each data point. **b**. Correlation of expression fold changes (log₂) of RNA seq data in m⁶A-depleted (STM2457, 24 h) over control conditions using normalisation to ERCC spike-ins (x-axis) or 100 randomly chosen genes without m⁶A sites (y-axis, see Methods; two-sided Pearson correlation coefficient [$R$] = 1, $P$ value < 2.2e-16). **c**. Upregulation upon m⁶A depletion increases with the number of m⁶A sites in the transcripts. Distribution of fold changes (log₂) in m⁶A-depleted (STM2457, 24 h) over control conditions in expressed transcripts (transcripts per million [TPM] > 1, based on total miCLIP2 signal) stratified by their number

of m⁶A sites. Numbers of transcripts in each category are indicated above. Boxes represent quartiles, centre lines denote medians, and whiskers extend to most extreme values within 1.5× interquartile range. **d**. Cumulative distribution of expressed autosomal (grey) and X-chromosomal (orange) transcripts (RPKM > 1) with a given expression level (RPKM, x-axis). The expression distributions of X-chromosomal and autosomal transcripts are largely identical, supporting a X:A ratio close to 1 across the full expression range. For comparison, a theoretical doubling of the X-chromosomal expression is shown (orange, dotted) which would exceed autosomal expression levels. **e**. Median X-to-autosome (X:A) expression ratios increase with higher RPKM cut-offs (>0, $n$ [genes] = 26,291, ≥0.25, $n$ = 13,795, ≥0.5, $n$ = 12,255, ≥1, $n$ = 10,849). Median X:A ratios for male mESC and 95% confidence intervals are given.

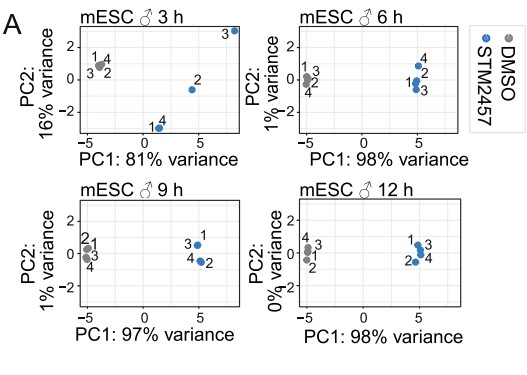

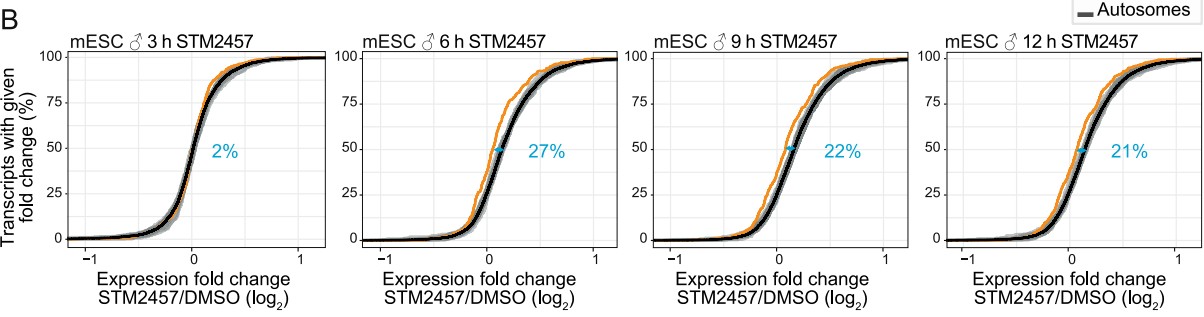

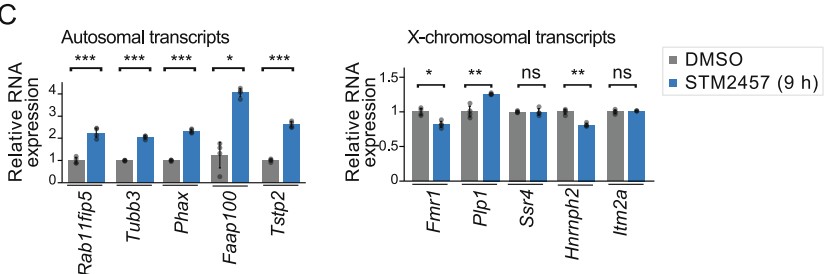

**Extended Data Fig. 4 | Time-course RNA-seq upon m⁶A depletion reveals upregulation of autosomal genes after 6 h of inhibitor treatment. a**. Principal component analyses of RNA-seq replicates of control and m⁶A-depleted male mESC at different time points (STM2457, 3–12 h) based on numbers of reads or the 500 genes with highest variance across all samples for a given time point. Replicate number given next to each data point. **b**. After 6 h of m⁶A depletion, X-chromosomal transcripts show significantly lower fold changes (log₂) compared to autosomal transcripts (*P* value = 0.48 [3 h], *P* value = 1.02e-12 [6 h], *P* value = 5.12e-10 [9 h], *P* value = 1.69e-08 [12 h], two-sided Wilcoxon rank-sum test). Cumulative fraction of transcripts on individual autosomes (grey) and the X chromosome (orange) that show a given expression fold change (log₂, RNA-seq) at different timepoints of m⁶A depletion (STM2457, 3–12 h) in male mESC. Mean expression changes for all autosomes are shown as black line. Effect sizes (blue) show the shift in medians, expressed as percent of the average interquartile range (IQR) of autosomal and X-chromosomal genes (see Methods). **c**. qPCR to quantify expression changes of five autosomal (left) and five X-chromosomal (right) transcripts in control and m⁶A-depleted (STM2457, 9 h) male mESC cells. Normalised $C_T$ values ($\Delta C_T$, normalised to *Gapdh* expression) are compared between conditions. Fold changes are displayed as mean ± s.d.m., two-sided Student's *t*-test on log₂-transformed data, *n* = 4 biologically independent samples, *\*P* value < 0.05, *\*\*P* value < 0.01, *\*\*\*P* value < 0.001, ns, not significant. *P* value = 0.00017 [*Rab11fip5*], 8.57e-07 [*Tubb3*], 8.08e-08 [*Phax*], 0.049 [*Faap100*], 1.46e-06 [*Tstp2*]; 0.56 [*Itm2a*], 0.001 [*Hnrnph2*], 0.95 [*Ssr4*], 0.007 [*Plp1*], 0.01 [*Fmr1*].

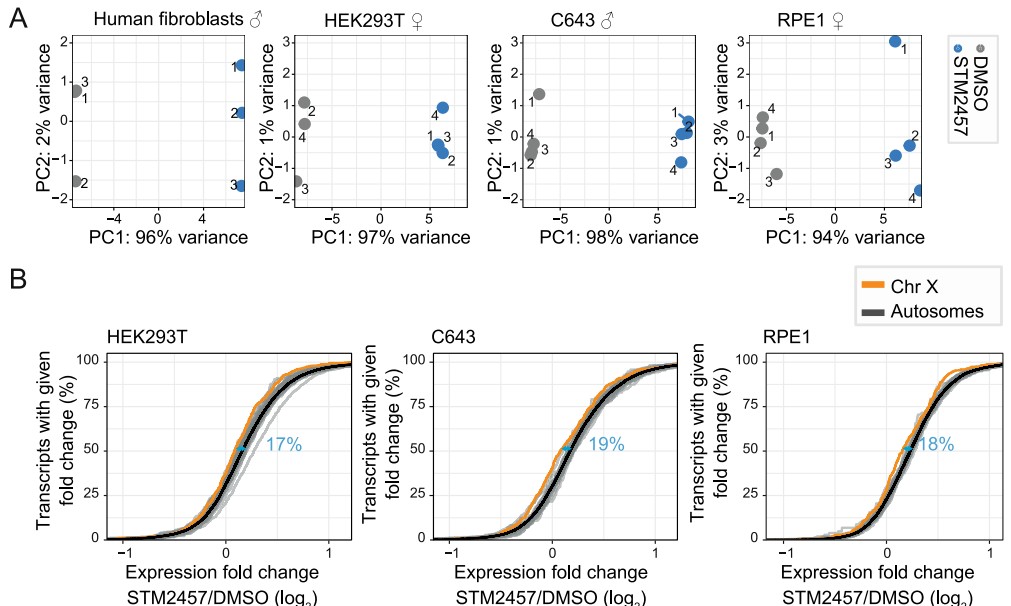

**Extended Data Fig. 5 | RNA-seq upon m⁶A depletion reveals upregulation of autosomal transcripts in human cell lines. a**. Principal component analyses for replicates of RNA-seq experiments under m⁶A-depleted and control conditions for human primary fibroblasts (STM2457, 9 h), HEK293T cells, C643 cells and RPE1 cells (STM2457, 24 h). Replicate number given next to each data point. **b**. X-chromosomal transcripts show significantly lower fold changes upon m⁶A depletion than autosomal transcripts ($P$ value = 6.92e-06 [HEK293T, $n$ = 12,856 of autosomal transcripts, $n$ = 443 of X-chromosomal transcripts], $P$ value = 4.53e-05 [C643, $n$ = 11,109 of autosomal transcripts, $n$ = 383 of X-chromosomal

transcripts], $P$ value = 0.0001901 [RPE1, $n$ = 10,732 of autosomal transcripts, $n$ = 347 of X-chromosomal transcripts], Wilcoxon rank-sum test). Cumulative fraction of transcripts on individual autosomes (grey) and the X chromosome (orange) that show a given fold change (log₂) in m⁶A-depleted (STM2457, 24 h) over control conditions for HEK293T, C643, and RPE1 cells. Mean expression changes for all autosomes are shown as black line. Effect sizes (blue) shown the shift in medians, expressed as percent of the average IQR of autosomal and X-chromosomal genes (see Methods).

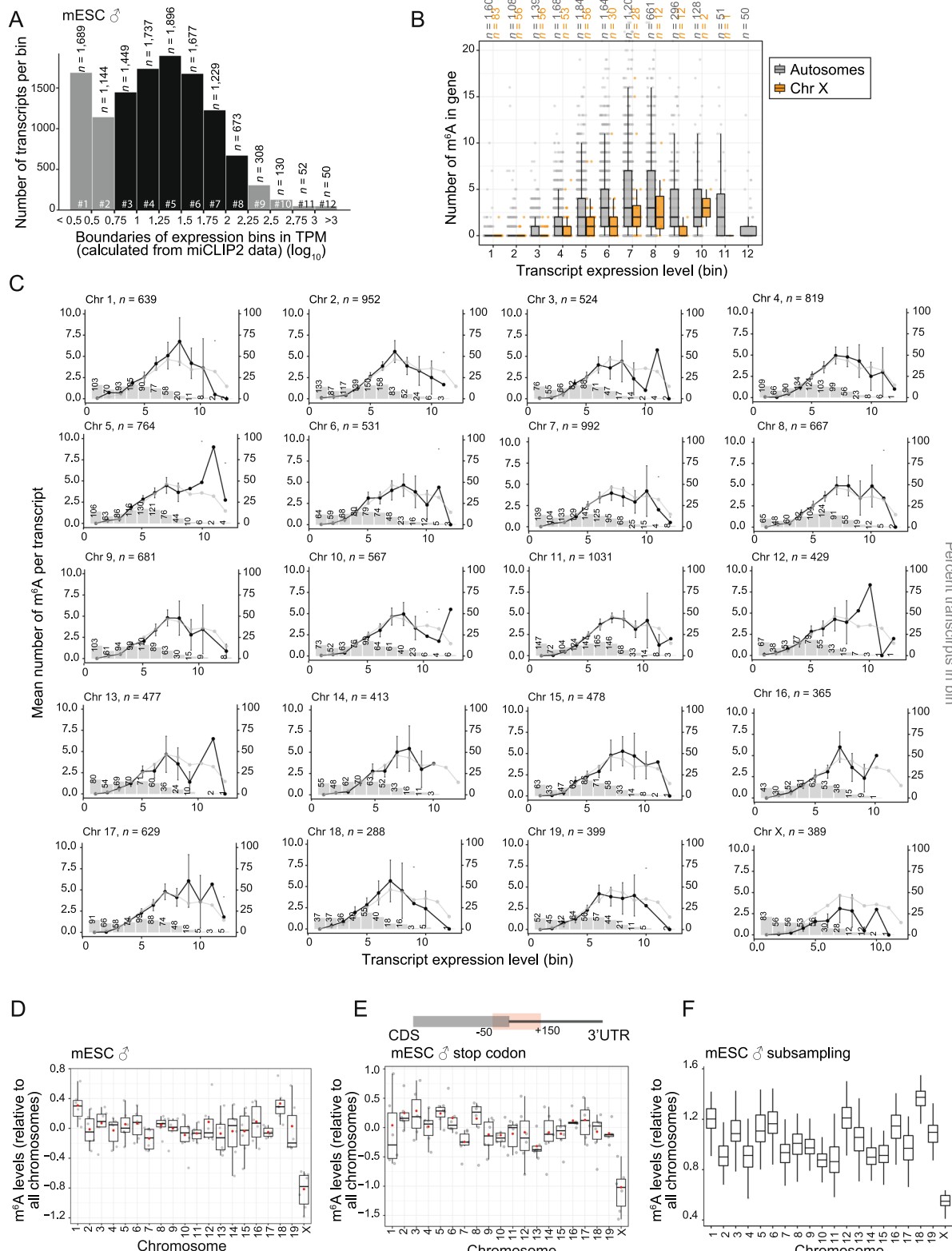

Extended Data Fig. 6 | See next page for caption.

**Extended Data Fig. 6 | X-chromosomal transcripts harbour less m⁶A sites than autosomal transcripts in male mESC. a**. Transcripts were stratified into 12 bins (#1–12) according to their expression in male mESC (transcripts per million [TPM, $\log_{10}$], see Methods). x-axis depicts boundaries between bins (in TPM). Bin number (#) and number of transcripts therein are given below and above each bar, respectively. Bins #3–8 that were used for quantifications of m⁶A sites per transcripts are highlighted in black. **b**. Quantification of m⁶A for each transcript in the different expression bins of autosomal (grey) and X-chromosomal (orange) transcripts. Boxes represent quartiles, centre lines denote medians, and whiskers extend to most extreme values within 1.5× interquartile range. **c**. Quantification of m⁶A sites per transcript for all mouse chromosomes. Data points indicate mean number of m⁶A sites per transcript and 95% confidence interval (left y-axis) in each expression bin (x-axis, bins as defined in **a**) for all chromosomes (chromosome name and total number of expressed transcripts given above).

Grey bars indicate the percentage of transcripts in each expression bin (right y-axis) relative to all expressed transcripts on the chromosome. Absolute numbers of transcripts in each bin are given above the bars. Only genes with a mean TPM > 1 over all samples were considered. **d**. Fold change ($\log_2$, grey dots) in mean m⁶A sites per transcripts for expression bins #3–8 ($n$ of mean of expression bins = 6) on an individual chromosome over the mean m⁶A sites per transcripts across all chromosomes. Red dots indicate mean fold change of the six bins on the given chromosome. Boxes as in **b**. **e**. Same as **d**. using only m⁶A sites in a fixed window around stop codons (−50 nt to +150 nt) to exclude confounding effects of transcript length differences. Boxes as in **b**. **f**. Same as **c**. after randomly subsampling $n$ = 30 genes from expression bins #3–5 to exclude potential biases from different numbers of transcripts in the expression bins for each chromosome. Shown is the distribution of mean m⁶A sites per transcript for each chromosome from 100 repeats of subsampling. Boxes as in **b**.

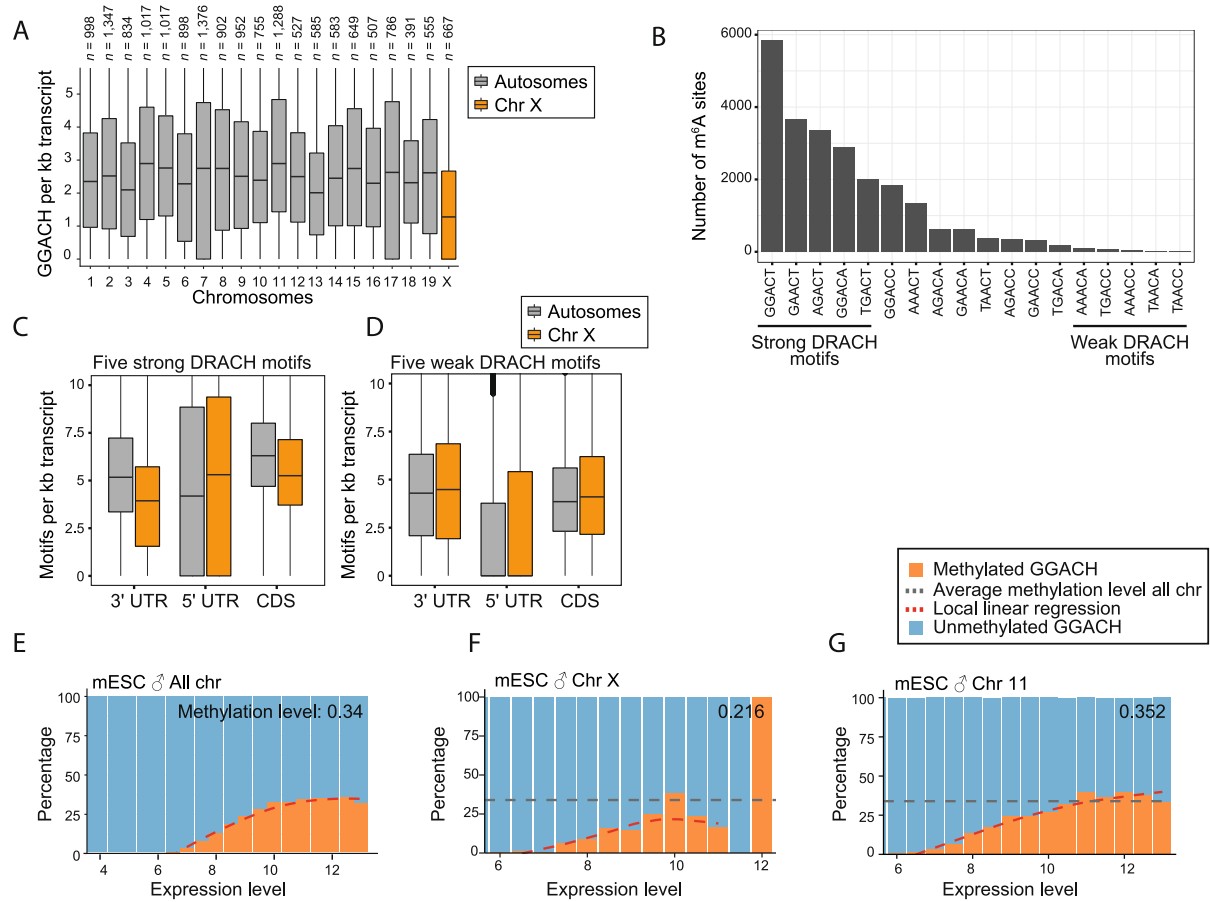

**Extended Data Fig. 7 | The number of GGACH motifs and their methylation level are reduced on X-chromosomal transcripts compared to autosomal transcripts. a**. X-chromosomal transcripts harbour fewer GGACH motifs than autosomal transcripts. Distribution of GGACH (H=[A|C|U]) per kilobase (kb) transcript sequence for individual chromosomes (corresponding to Fig. 4a). Boxes represent quartiles, centre lines denote medians, and whiskers extend to most extreme values within 1.5× interquartile range. **b**. Distribution of m⁶A sites from mESC miCLIP2 data across different DRACH motifs. Barplot shows the number of m⁶A sites for a given type of DRACH motif in mESC. The five most often methylated ('strong') and least often methylated ('weak') DRACH motifs are labelled below. **c**. Autosomal transcripts harbour more frequently methylated DRACH motifs in CDS and 3' UTR. Quantification of strong DRACH motifs in different transcript regions (normalised to region length) of autosomal (grey) and X-chromosomal transcripts (orange) in mouse. CDS $n$ of annotations = 16,631, 3' UTR $n$ of annotations = 16,484 and 5' UTR $n$ of annotations = 16,490.

Boxes as in **a. d**. Autosomal transcripts harbour similar numbers of the least methylated DRACH motifs ('weak') in CDS and 3' UTR. Quantification of the five least methylated DRACH motifs as in *(C.)*. CDS $n$ of annotations = 16,631, 3' UTR $n$ of annotations = 16,484 and 5' UTR $n$ of annotations = 16,490. Boxes as in **a. e–g**. The methylation level of GGACH motifs in male mESC, that is, the percentage of GGACH motifs that are methylated, is slightly reduced in X-chromosomal transcripts (**f**), compared to transcripts across all chromosomes (**e**) or from chromosome 11 (**g**). To take into account only GGACH motifs in transcript regions with sufficient expression, GGACH motifs in transcripts were stratified into bins by the local miCLIP2 read coverage (see Methods) and overlayed with m⁶Aboost-predicted m⁶A sites from the same data. Dashed red line indicates local linear regression to estimate the methylation level (shown in Fig. 4b), that is, the point at which the slope drops below 0.01. Dashed grey lines in **f** and **g** show estimated GGACH methylation level for transcripts across all chromosomes (**e**) for comparison.

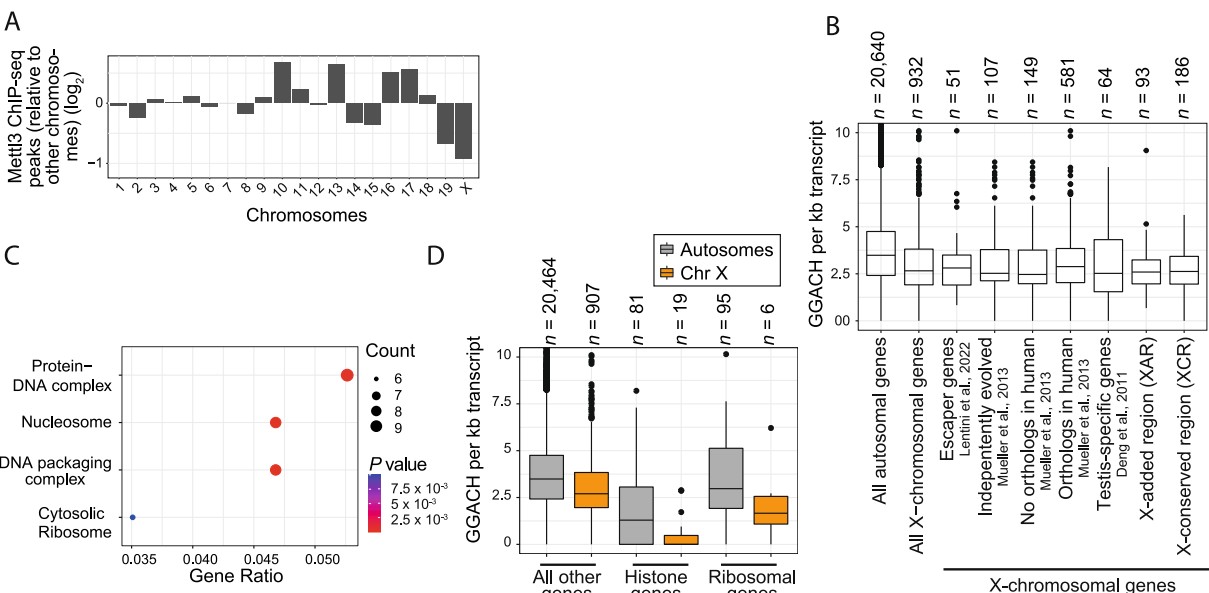

**Extended Data Fig. 8 | The number of GGACH motifs is reduced on transcripts encoding histones and ribosomal proteins. a**. The X chromosome harbours fewer Mettl3 ChIP-seq peaks. The number of published ChIP-seq peaks (normalised by chromosome length) per chromosome relative to peaks on all other chromosomes ($\log_2$). **b**. Different gene sets on the X-chromosome are similarly depleted in GGACH motifs. Quantification of GGACH motifs of all autosomal or X-chromosomal genes is compared to the following gene sets: escaper genes, independently evolved genes, genes with or without orthologs on the human X chromosome, testis-specific genes or genes residing in the X-added region (XAR) and X-conserved region (XCR). Numbers of genes are given in the

figure ($n$). Boxes represent quartiles, centre lines denote medians, and whiskers extend to most extreme values within 1.5× interquartile range. **c**. X-chromosomal genes with low GGACH motif numbers are associated with DNA packaging or the cytosolic ribosome. Gene ontology (GO) enrichment analysis of the 200 genes with the lowest density of GGACH motifs on the X chromosome. $P$ values were calculated by overrepresentation analysis (see Methods). **d**. Histone and ribosomal protein-encoding genes on the X chromosome are depleted in GGACH motifs. Quantification of GGACH motifs for histone-encoding and ribosomal protein-encoding genes on autosomes or on the X chromosome. Numbers of genes are given in the figure ($n$). Boxes as in **b**.

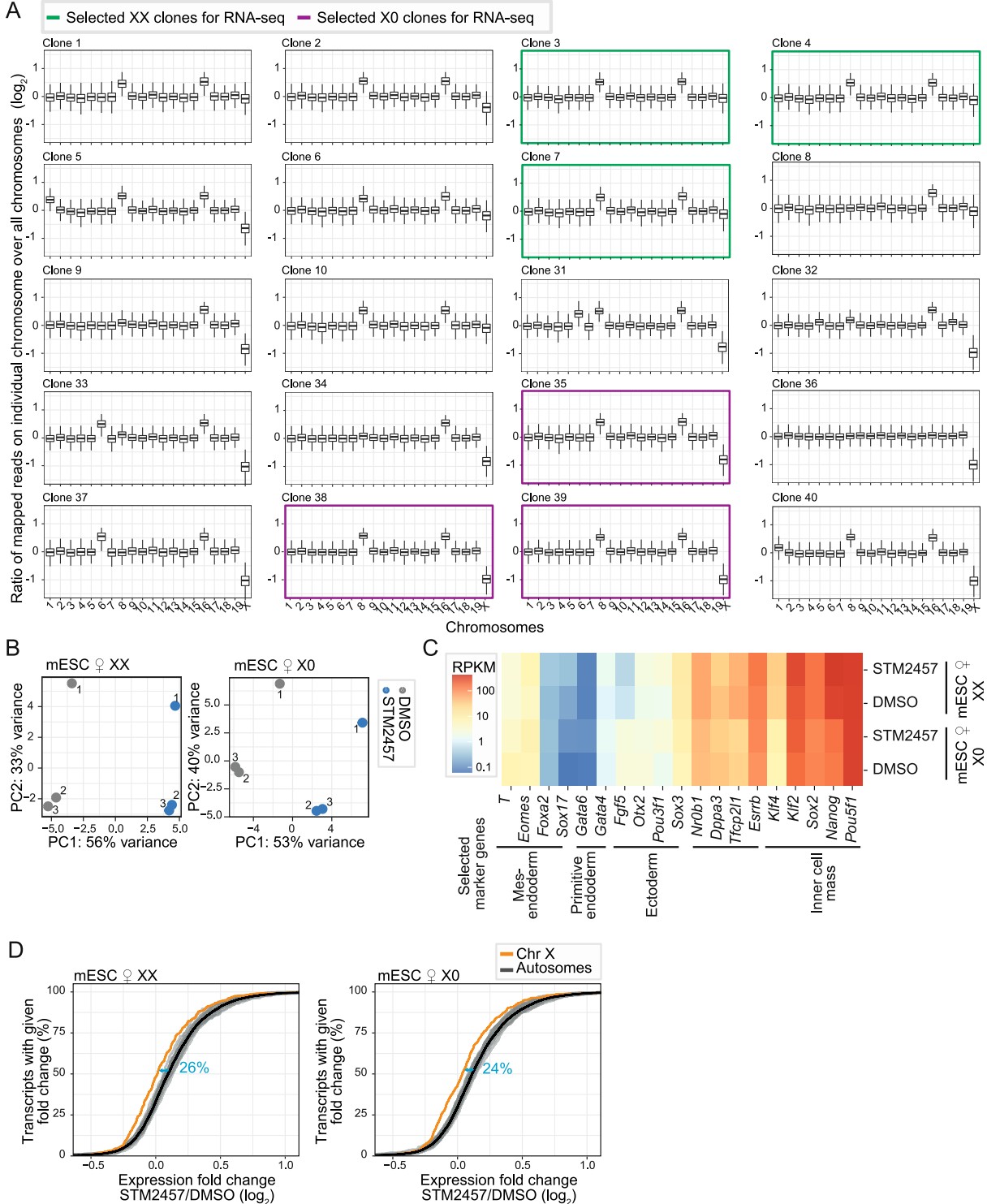

**Extended Data Fig. 9 | X-chromosomal and autosomal transcripts differ in their response to m⁶A depletion in both XX or X0 clones of female mESC.**
**a**. The majority of clones lost one copy of the X chromosome (X0). 20 single colonies of female mESC were picked and cultured under standard conditions until confluency was reached. To determine chromosome copy number, DNA-seq reads were counted into 100 kb bins along the chromosome and divided by the median mapped reads of all bins along the genome. Shown is the distribution of the resulting ratios for the bins on each chromosome. Six clones that were selected for RNA-seq in control and m⁶A-depleted (STM2457, 9 h) condition are highlighted in green. Boxes represent quartiles, centre lines denote 50th percentiles (medians), and whiskers extend to most extreme values within 1.5× interquartile range. **b**. Principal component analysis of RNA-seq replicates from female X0 (left) and XX (right mESC clones under m⁶A-depleted (STM2457, 9 h)

and control conditions. Analysis based on numbers of reads for the 500 genes with highest variance across all samples. **c**. Expression levels (RNA-seq) of marker genes confirm the pluripotent state of the female XX and X0 mESC under m⁶A-depleted (STM2457, 9 h) and control conditions. Expression is shown as RPKM (mean over replicates, log₁₀). **d**. X-chromosomal transcripts are less upregulated than autosomal transcripts upon m⁶A depletion in female X0 and XX mESC (P value = 3.51e-11 [mESC X0], P value = 1.64e-12 [mESC XX], two-sided Wilcoxon rank-sum test). Cumulative fraction of transcripts (RPKM > 1) on individual autosomes (grey) and the X chromosome (orange) that show a given expression fold change (log₂, RNA-seq) upon m⁶A depletion (STM2457, 9 h). Mean expression changes for all autosomes are shown as black line. Effect sizes (blue) shown the shift in medians, expressed as percent of the average IQR of autosomal and X-chromosomal transcripts (see Methods).

# Reporting Summary

## Statistics

For all statistical analyses, confirm that the following items are present in the figure legend, table legend, main text, or Methods section.

| n/a | Confirmed | |
|---|---|---|
| ☐ | ☒ | The exact sample size (*n*) for each experimental group/condition, given as a discrete number and unit of measurement |
| ☐ | ☒ | A statement on whether measurements were taken from distinct samples or whether the same sample was measured repeatedly |
| ☐ | ☒ | The statistical test(s) used AND whether they are one- or two-sided<br>*Only common tests should be described solely by name; describe more complex techniques in the Methods section.* |
| ☒ | ☐ | A description of all covariates tested |
| ☐ | ☒ | A description of any assumptions or corrections, such as tests of normality and adjustment for multiple comparisons |
| ☐ | ☒ | A full description of the statistical parameters including central tendency (e.g. means) or other basic estimates (e.g. regression coefficient) AND variation (e.g. standard deviation) or associated estimates of uncertainty (e.g. confidence intervals) |
| ☐ | ☒ | For null hypothesis testing, the test statistic (e.g. *F*, *t*, *r*) with confidence intervals, effect sizes, degrees of freedom and *P* value noted<br>*Give P values as exact values whenever suitable.* |
| ☒ | ☐ | For Bayesian analysis, information on the choice of priors and Markov chain Monte Carlo settings |
| ☒ | ☐ | For hierarchical and complex designs, identification of the appropriate level for tests and full reporting of outcomes |
| ☐ | ☒ | Estimates of effect sizes (e.g. Cohen's *d*, Pearson's *r*), indicating how they were calculated |

*Our web collection on statistics for biologists contains articles on many of the points above.*

## Software and code

Policy information about availability of computer code

| Data collection | For the following datasets, the given tools with given versions were used.<br>SLAM-seq processing<br>FastQC (v0.11.8)<br>SLAM-DUNK (v0.4.3)<br>lme4 (v1.1.29)<br> lmerTest (v3.1.3)<br><br>RNA Seq processing<br>FastQC (v0.11.8)<br>Cutadapt (v1.18)<br>STAR (v2.7.3a)<br>SAMtools (v1.9)<br>DESeq2 (v1.26.0)<br>Subread tool suite (v2.0.0)<br><br>miCLIP2 processing<br>FastQC (v0.11.8)<br>Flexbar (v3.4.0)<br>STAR (v2.7.3a)<br>CTK, v1.1.3)<br>SAMtools (v1.9)<br>BEDTools v2.27.1<br>PureCLIP (version 1.3.1)<br>FASTX-Toolkit (v0.0.14) |
|---|---|

```
seqtk (v1.3)
bedGraphToBigWig of the UCSC tool suite (v365)

DNA seq Processing
FastQC (v0.11.8)
Cutadapt (v2.4)
Bowtie2 (v2.3.4.3)
SAMtools (v1.9)
Picard (v2.20.3)

Differential gene expression analysis
DESeq2 (v1.34.0)
lme4 (v1.1.29)
lmerTest (v3.1.3)
 emmeans (v1.8.0)

GO analysis
clusterProfiler (v.4.2.2)

Comparison of m6A sites
htseq-count80 (v0.11.1)
multcomp (v1.4.19)
Biostrings (v2.59.2)
bamCoverage (v3.5.1)
```

| Data analysis | All data analysis steps performed for this manuscript are described in detail in the manuscript. The scripts used to process the files are accessible under the GitHub repository located at: github.com/crueckle/Rueckle_et_al_2023 |
|---|---|

For manuscripts utilizing custom algorithms or software that are central to the research but not yet described in published literature, software must be made available to editors and reviewers. We strongly encourage code deposition in a community repository (e.g. GitHub). See the Nature Portfolio guidelines for submitting code & software for further information.

# Data

Policy information about availability of data

All manuscripts must include a data availability statement. This statement should provide the following information, where applicable:
- Accession codes, unique identifiers, or web links for publicly available datasets
- A description of any restrictions on data availability
- For clinical datasets or third party data, please ensure that the statement adheres to our policy

All high-throughput sequencing datasets generated in this study were submitted to the Gene Expression Omnibus (GEO) under the SuperSeries accession GSE203653 (https://www.ncbi.nlm.nih.gov/geo/query/acc.cgi?&acc=GSE203653). RNA-seq data for human primary fibroblasts is available upon request. HEK293T and C643 miCLIP data was taken from Gene Expression Omnibus with the accession number GSE163500.
m6A-seq2 data was taken from Gene Expression Omnibus with the accession number  GSE178832
ChIP-seq data was taken from Gene Expression Omnibus with the accession number GSE126243

# Field-specific reporting

Please select the one below that is the best fit for your research. If you are not sure, read the appropriate sections before making your selection.

☒ Life sciences          ☐ Behavioural & social sciences          ☐ Ecological, evolutionary & environmental sciences

For a reference copy of the document with all sections, see nature.com/documents/nr-reporting-summary-flat.pdf

# Life sciences study design

All studies must disclose on these points even when the disclosure is negative.

| Sample size | Samples sizes were chosen based on accepted practises in the field and stated in each figure or figure legend with at least three biological replicates. Example references from the field:<br>miCLIP2: Körtel N, Rücklé C, Zhou Y, et al. Deep and accurate detection of m6A RNA modifications using miCLIP2 and m6Aboost machine learning. Nucleic Acids Res. 2021;49(16):e92. doi:10.1093/nar/gkab485<br>RNA-seq: Yankova E, Blackaby W, Albertella M, et al. Small-molecule inhibition of METTL3 as a strategy against myeloid leukaemia. Nature. 2021;593(7860):597-601. doi:10.1038/s41586-021-03536-w<br>SLAM-seq: Herzog VA, Reichholf B, Neumann T, et al. Thiol-linked alkylation of RNA to assess expression dynamics. Nat Methods. 2017;14(12):1198-1204. doi:10.1038/nmeth.4435<br>Rothamel K, Arcos S, Kim B, et al. ELAVL1 primarily couples mRNA stability with the 3' UTRs of interferon-stimulated genes. Cell Rep. 2021;35(8):109178. doi:10.1016/j.celrep.2021.109178 |
|---|---|
| Data exclusions | For RNA-seq in HEK293T cells, one replicate of DMSO condition was exluded. Gene body coverage for this sample indicated RNA degradation |

| Data exclusions | prior to RNA-seq library preparation. Otherwise, no data was exluded. |
| Replication | Each experiment was performed in at least three replicates unless stated otherwise. All replictaion attempts were successfull. |
| Randomization | Samples were collected to study groups by genotype and/or treatment condition. No randomization was applied. |
| Blinding | Blinding was not relevant due to the objective readouts (e.g. RNA-seq, SLAM-seq, qPCR) and investigators were not blinded. Investigators were not blinded. No placebo effects were expected since no patients were involved. |

# Reporting for specific materials, systems and methods

We require information from authors about some types of materials, experimental systems and methods used in many studies. Here, indicate whether each material, system or method listed is relevant to your study. If you are not sure if a list item applies to your research, read the appropriate section before selecting a response.

## Materials & experimental systems

| n/a | Involved in the study |
|---|---|
| ☐ | ☒ Antibodies |
| ☐ | ☒ Eukaryotic cell lines |
| ☒ | ☐ Palaeontology and archaeology |
| ☒ | ☐ Animals and other organisms |
| ☐ | ☒ Human research participants |
| ☒ | ☐ Clinical data |
| ☒ | ☐ Dual use research of concern |

## Methods

| n/a | Involved in the study |
|---|---|
| ☒ | ☐ ChIP-seq |
| ☒ | ☐ Flow cytometry |
| ☒ | ☐ MRI-based neuroimaging |

## Antibodies

| Antibodies used | m6A antibody, Synaptic Systems (cat. number: 202 003) RRID:AB_2279214. 6 µg of antibody was used per 1 µg of input RNA per replicate. |
| Validation | The used m6A antibody was obtained from a commercial vendor which ensures the quality of the antibody. In a previous publication (PMID: 34157120) the antibody was validated for used application. |

## Eukaryotic cell lines

Policy information about cell lines

| Cell line source(s) | HEK293T were purchased from CSL (order number: Cryovial: 300192 Vital: 330192). C643 were purchased from CSL RRID:CVCL 5969. RPE1 cells were purchased from ATCC (order number: CRL-4000). Males mESC were provided by Dan Dominissini. Female mESC (TX1072) were provided by Edith Heard: TX1072 |
| Authentication | Cells were not authenticated after the purchase. |
| Mycoplasma contamination | All cell lines are monitored and tested negative for Mycoplasma contamination. |
| Commonly misidentified lines (See ICLAC register) | No commonly misidentified cell lines were used in this study. |

## Human research participants

Policy information about studies involving human research participants

| Population characteristics | The donor of human fibroblasts was a healthy 25 year old male proband. |
| Recruitment | The recruitment as control participants in a study on neurodegenerative diseases. |
| Ethics oversight | Ethical approval by the local ethical committee was obtained (No. 4485), and consent for research use in an anonymised way was given. The ethical approval given by the ethical committee of the University medicine at the Johannes Gutenberg University in Mainz, Germany, No. 4485 |

Note that full information on the approval of the study protocol must also be provided in the manuscript.

