## [Peer Review File · Nature Structural & Molecular Biology]

Peer Review Information

Manuscript Title: RNA stability controlled by m⁶A methylation contributes to X-to-autosome dosage compensation in mammals

Corresponding author name(s): Julian König

Editorial Notes:

Redactions – transferred manuscripts (mention of previous referee reports from elsewhere) This manuscript has been previously reviewed at another journal. This document only contains reviewer comments, rebuttal and decision letters for versions considered at Nature Structural & Molecular Biology. Mentions of prior referee reports have been redacted

Reviewer Comments & Decisions:

Decision Letter, initial version:

Message: Our ref: NSMB-A46934-T

2nd Jan 2023

Dear Dr. König,

Thank you for submitting your revised manuscript "RNA stability controlled by m⁶A methylation mediates X-to-autosome dosage compensation in mammals" (NSMB-A46934-T). I sincerely apologize for the delay in responding due to a significantly reduced editorial capacity over the holidays. The manuscript has now been seen by the original referees and their comments are below. The reviewers find that the paper has improved in revision, and therefore we'll be happy in principle to publish it in Nature Structural & Molecular Biology, pending revisions to satisfy the referees' final requests and to comply with our editorial and formatting guidelines.

We are now performing detailed checks on your paper and will send you a checklist detailing our editorial and formatting requirements in about two weeks. Please do not upload the final materials and make any revisions until you receive this additional information from us.

To facilitate our work at this stage, we would appreciate if you could send us the main text as a word file. Please make sure to copy the NSMB account (cc'ed above).

Sincerely,
Sara

Sara Osman, Ph.D.
Associate Editor
Nature Structural & Molecular Biology

Reviewer #1 (Remarks to the Author):

The manuscript seeks to explain why X chromosomes (which is present in one copy) has gene expression that matches autosomal chromosomes (which are present as 2 copies). They say that autosomal genes are more likely to have m6A.

I think this is improved. The more clearly quantify the effect of m6A and show that the effect is real, but the revision shows that the effect is small. So, m6A contributes to, but does not explain the gene expression difference. I think this a more accurate story. Since the magnitude of the effect is small the story could be more appropriate for Nat Comm, but is also borderline for NSMB. Regardless, the current version is better since it gives a clearer assessment of the role of m6A in dosage compensation, even if it is small.

Comments

1.Rev 1 and 3 were both concerned that the authors imply that m6A was the major cause of autosome gene suppression that causes autosomes to have low expression that matches the expression from the X chromosome. The new experiments with X:A ratios are very useful to show that m6A only -partly- contributes to this effect. The authors say they have now made it more clear that m6A only contributes partly to this phenomenon. But the title is still misleading (Rev 3 said it should be changed). I suggest they change:

RNA stability controlled by m6A methylation mediates X-to-autosome dosage compensation in mammals

to

RNA stability controlled by m6A *contributes* to X-to-autosome dosage compensation in mammals

The first title is very misleading. The revised title is accurate.

2.One concern was the emphasis on GGACH as the m6A motif, which is not completely correct as the m6A motif. The rebuttal shows that the second most common m6A motif is GAACU and the third is AGACU. These are not GGACH sequenes. So, the authors themselves show that focusing on GGACH in the main figures is not valid. It is good that

they redid some analysis using the correct motifs, or at least the top five m6A motifs in the new Supplementary Fig. 6B, C and D. Since this is the -correct- motif, S6C and D should be in the main figures. I would prefer that they look at the top five motifs, not GGACH for -all their panels- where they looked at GGACH, e.g. S6I-K etc. The authors should be looking for the top five m6A motifs, not a single m6A motif GGACH. This is simply bioinformatic and should not require any bench work so I don't think this is an onerous request. I think readers will be confused about the fixation with GGACH rather than the top m6A motifs.

3. The new X:A ratio data is great and really helps to show that m6A has a real but -small- effect on explaining the dosage compensation of the X chromosome. But this needs to be in the main figures - people must see this to see that the role of m6A is real but small. So figures such as S3F, S7D and S4C should be main figures since these are really the -best and most convincing- experiments, even if they show a small effect of m6A inhibition.

Importantly, the authors have used a misleading y-axis. The y-axis should go from 0.5 to 1.0 since the X could have 50% to 100% of the expression of the autosome. But making the y-axis go from 0.8 to 1.0, the small effect of the m6A pathway inhibitors looks a lot more pronounced. An accurate y-axis is important to make sure no one is misled. If the authors think the readers won't be able to see the effect using an axis from 0.5 to 1.0+, they can use a box and magnify the area of interest on the graph as an adjacent image. But it is important to see this data and the magnitude of the effect using a correct y-axis.

Reviewer #2 (Remarks to the Author):

The reviewers have addressed in great detail all of my comments and have produced an excellent final product. I feel compelled to advocate on behalf of the authors with respect to Reviewer 1. That reviewer states "the idea that genes on autosomes are expressed less efficiently because they contain m6A is a flawed concept. Only a small fraction of transcripts have sufficient levels of m6A to be meaningfully regulated by the m6A pathway". That reviewer's comment shows a lack of understanding of X chromosome upregulation. X-upregulation is not predicted to affect all transcripts; indeed it has been argued that tissue-specific autosomal and X-transcripts do not need to be balanced, because imbalances would not affect overall fitness. So I think that reviewer's comment is incorrect. I note that the authors have also presented their own, well-reasoned response to this comment.

Reviewer #3 (Remarks to the Author):

In this revised version of their interesting manuscript the authors should be commended for their useful additional analyses addressing many of the comments from the three reviewers. However, while the authors generally toned down their claim that m6A RNA methylation was THE mechanism of dosage compensation, parts of the paper have not been appropriately changed and thus additional clarifications are needed as described below:

1. The title should read: "RNA stability controlled by m6A methylation contributes to X-to-autosome dosage compensation in mammals". Indeed, as stated by the authors line171,

"m6A acts in addition to other regulatory mechanisms...". Similarly, in the Abstract, line 35 should read: "...dosage compensation is partly regulated by epitranscriptomic RNA modification."

2. In the Discussion, line 337 should read: "...thereby affects the X-to-autosome balance of gene expression", and line 351 should read: "...thereby partly disrupting the X-to-autosome dosage compensation."

In the legend of Figure 4, lines 983 and 989 should read: "...contributes to X-to-autosome dosage compensation." The depiction of what I presume is the Y chromosome in Figure 4 is strange, why is there only one arm on one side of the centromere?

3. Description of the analysis of the opossum data is incomplete. In the text, line 282, there should be a description of the findings in opossum with a distinction between XCR (X conserved region) and XAR (X added region) regions of the eutherian X chromosome (see reference 1). The XCR contains genes on the X chromosome in both eutherian mammals and marsupials, whereas the XAR is X-linked only in eutherian mammals and autosomal in marsupials. Figure 3 and its legend (line 955) should also include a distinction between the XCR and the XAR to provide a measure of GGACH motifs abundance (Fig. 3E) for the X-linked genes of opossum corresponding to the XCR and for the autosomal genes of opossum corresponding to the XAR. In addition, the method section does not include analysis of the opossum data: see lines 746 and 784.

4. Line 261-263: The authors should include a statement about escape genes shown in Fig S6I to indicate that they behave similarly to other X-linked genes. If this is the case this would contradict the statement that escape genes may not need compensation (line 366 in the discussion). This should be clarified.

Author Rebuttal to Initial comments

Point by point response

Reviewer #1:

Remarks to the Author:

The manuscript seeks to explain why X chromosomes (which is present in one copy) has gene expression that matches autosomal chromosomes (which are present as 2 copies). They say that autosomal genes are more likely to have m6A.

I think this is improved. The more clearly quantify the effect of m6A and show that the effect is real, but the revision shows that the effect is small. So, m6A contributes to, but does not explain the gene expression difference. I think this a more accurate story. Since the magnitude of the effect is small the story could be more appropriate for Nat Comm, but is also borderline for NSMB. Regardless, the current version is better since it gives a clearer assessment of the role of m6A in dosage compensation, even if it is small.

Comments

1.Rev 1 and 3 were both concerned that the authors imply that m6A was the major cause of autosome gene suppression that causes autosomes to have low expression that matches the expression from the X chromosome. The new experiments with X:A ratios are very useful to show that m6A only -partly- contributes to this effect. The authors say they have now made it more clear that m6A only contributes partly to this phenomenon. But the title is still misleading (Rev 3 said it should be changed). I suggest they change:

RNA stability controlled by m6A methylation mediates X-to-autosome dosage compensation in mammals

to

RNA stability controlled by m6A *contributes* to X-to-autosome dosage compensation in mammals

The first title is very misleading. The revised title is accurate.

We have changed the title accordingly.

2.One concern was the emphasis on GGACH as the m6A motif, which is not completely correct as the m6A motif. The rebuttal shows that the second most common m6A motif is GAACU and the third is AGACU. These are not GGACH sequences. So, the authors themselves show that focusing on GGACH in the main figures is not valid. It is good that they redid some analysis using the correct motifs, or at least the top five m6A motifs in the new Supplementary Fig. 6B, C and D. Since this is the -correct- motif, S6C and D should be in the main figures. I would prefer that they look at the top five motifs, not GGACH for -all their panels- where they looked at GGACH, e.g. S6I-K etc. The authors should be looking for the top five m6A motifs, not a single m6A motif GGACH. This is simply bioinformatic and should not require any bench work so I don't think this is an onerous request. I think readers will be confused about the fixation with GGACH rather than the top m6A motifs.

While the DRACH consensus comprises several different motifs, it is known that some of these motifs are more often methylated than others. We have intentionally focused on GGACH motifs, since it was shown by several publications that GGACU is the most commonly methylated motif (Linder et al., 2015, *Nature Methods*; Pratanwanich et al., 2021, *Nature Biotechnology*) and therefore several studies focused on GGACH motifs since it most likely reflects the m⁶A deposition well (Xiong et al., 2021, *Nature Genetics*; Sun et al., 2021, *Nature Communications*; Visvanathan et al., 2017, *Oncogene*). We had therefore used GGACH in all our initial analyses for this manuscript. In response to the first round of reviewers' comments, we included in our revised version the additional analysis in Supplementary Figure S6B-D to show that this result (reduced m⁶A motifs in X-chromosomal transcripts) can be reproduced with a different subset of DRACH motifs, i.e. the five most frequently methylated DRACH motifs in our miCLIP2 data. We think that

adding the top 5 motifs in the revision process has enhanced our manuscript and served as a nice control. However, we politely disagree to change all our main analyses to this motif set. We think that this set of motifs is not necessarily a better reflection of m⁶A deposition, since (i) the m⁶A site counts are as such not seen in relation to the underlying abundance of these motifs in transcript regions, and (ii) the trailing U in the two mentioned DRACH variants could to a certain extent be favoured in the miCLIP2 data due to the increased UV crosslinking activity of uridines. Based on these considerations, we decided to keep the main analyses in the manuscript unchanged.

3. The new X:A ratio data is great and really helps to show that m6A has a real but -small- effect on explaining the dosage compensation of the X chromosome. But this needs to be in the main figures - people must see this to see that the role of m6A is real but small. So figures such as S3F, S7D and S4C should be main figures since these are really the -best and most convincing- experiments, even if they show a small effect of m6A inhibition.

We have moved the respective analyses into the main figures (now Figure 2B, 2E, 4H).

Importantly, the authors have used a misleading y-axis. The y-axis should go from 0.5 to 1.0 since the X could have 50% to 100% of the expression of the autosome. But making the y-axis go from 0.8 to 1.0, the small effect of the m6A pathway inhibitors looks a lot more pronounced. An accurate y-axis is important to make sure no one is misled. If the authors think the readers won't be able to see the effect using an axis from 0.5 to 1.0+, they can use a box and magnify the area of interest on the graph as an adjacent image. But it is important to see this data and the magnitude of the effect using a correct y-axis.

We have changed the y-axis accordingly.

Reviewer #2:

Remarks to the Author:

The reviewers have addressed in great detail all of my comments and have produced an excellent final product. I feel compelled to advocate on behalf of the authors with respect to Reviewer 1. That reviewer states "the idea that genes on autosomes are expressed less efficiently because they contain m6A is a flawed concept. Only a small fraction of transcripts have sufficient levels of m6A to be meaningfully regulated by the m6A pathway". That reviewer's comment shows a lack of understanding of X chromosome upregulation. X-upregulation is not predicted to affect all transcripts; indeed it has been argued that tissue-specific autosomal and X-transcripts do not need to be balanced, because imbalances would not affect overall fitness. So I think that reviewer's comment is incorrect. I note that the authors have also presented their own, well-reasoned response to this comment.

We thank the reviewer for the appreciation and support.

Reviewer #3:

Remarks to the Author:

In this revised version of their interesting manuscript the authors should be commended for their useful additional analyses addressing many of the comments from the three reviewers. However, while the authors generally toned down their claim that m6A RNA methylation was THE mechanism of dosage compensation, parts of the paper have not been appropriately changed and thus additional clarifications are needed as described below:

1. The title should read: “RNA stability controlled by m6A methylation contributes to X-to-autosome dosage compensation in mammals”. Indeed, as stated by the authors line171, “m6A acts in addition to other regulatory mechanisms...”. Similarly, in the Abstract, line 35 should read: “...dosage compensation is partly regulated by epitranscriptomic RNA modification.”

We have changed the title and text accordingly. We have changed the abstract accordingly.

2. In the Discussion, line 337 should read: “...thereby affects the X-to-autosome balance of gene expression”, and line 351 should read: “...thereby partly disrupting the X-to-autosome dosage compensation.”

In the legend of Figure 4, lines 983 and 989 should read: “...contributes to X-to-autosome dosage compensation.” The depiction of what I presume is the Y chromosome in Figure 4 is strange, why is there only one arm on one side of the centromere?

We have changed the respective lines in the manuscript. We have also updated Figure 4 (now Figure 5) accordingly.

3. Description of the analysis of the opossum data is incomplete. In the text, line 282, there should be a description of the findings in opossum with a distinction between XCR (X conserved region) and XAR (X added region) regions of the eutherian X chromosome (see reference 1). The XCR contains genes on the X chromosome in both eutherian mammals and marsupials, whereas the XAR is X-linked only in eutherian mammals and autosomal in marsupials. Figure 3 and its legend (line 955) should also include a distinction between the XCR and the XAR to provide a measure of GGACH motifs abundance (Fig. 3E) for the X-linked genes of opossum corresponding to the XCR and for the autosomal genes of opossum corresponding to the XAR. In addition, the method section does not include analysis of the opossum data: see lines 746 and 784.

We apologise that the reviewer did not find the details on the opossum data analysis in the methods section, even though they had been present. We updated the respective subheading to increase findability (“GGACH motifs in chicken, opossum and human orthologs”). Regarding the abundance of

m⁶A sites in XCR- and XAR-associated genes in opossum, we refrained from a more detailed analysis at this point. It is important to note that m⁶A sites have not been directly mapped in opossum yet, so any analyses would remain preliminary at this point. We added the following sentences to point out this interesting direction for future studies:

Line 287-291: It will be interesting to generate m⁶A maps in different mammalian species to disentangle the contribution of m⁶A to the evolution of mammalian dosage compensation. This will also enable the investigation of X-chromosomal regions of different evolutionary origin such as X-added region (XAR), X-conserved region (XCR) and pseudoautosomal region (PAR).

4. Line261-263: The authors should include a statement about escape genes shown in Fig S6I to indicate that they behave similarly to other X-linked genes. If this is the case this would contradict the statement that escape genes may not need compensation (line 366 in the discussion). This should be clarified.

We added a description of the GGACH depletions in escaper genes:

Line 262-265: Furthermore, X-chromosomal genes that have been reported to escape X chromosome inactivation (escaper genes) did not show a significant difference in GGACH motifs, suggesting that they are equally depleted in m⁶A sites as other X-chromosomal genes⁴⁰.

We removed the sentence in the discussion:

Line 366: Consistently, several genes requiring the full two-fold upregulation escape X chromosome inactivation and hence are not in need for X-to-autosome dosage compensation⁵⁶

Final Decision Letter:

Message 6th Apr 2023

:

Dear Dr. König,

We are now happy to accept your revised paper "RNA stability controlled by m⁶A methylation contributes to X-to-autosome dosage compensation in mammals" for publication as a Article in Nature Structural & Molecular Biology.

Your paper will be published online soon after we receive proof corrections and will appear in print in the next available issue. You can find out your date of online publication by contacting the production team shortly after sending your proof corrections. Content is published online weekly on Mondays and Thursdays, and the embargo is set at 16:00 London time (GMT)/11:00 am US Eastern time (EST) on the day of publication. Now is the time to inform your Public Relations or Press Office about your paper, as they might be interested in promoting its publication. This will allow them time to prepare an accurate and satisfactory press release. Include your manuscript tracking number (NSMB-A46934A) and our journal name, which they will need when they contact our press office.

About one week before your paper is published online, we shall be distributing a press release to news organizations worldwide, which may very well include details of your work. We are happy for your institution or funding agency to prepare its own press release, but it must mention the embargo date and Nature Structural & Molecular Biology. If you or your Press Office have any enquiries in the meantime, please contact press@nature.com.

If you have not already done so, we strongly recommend that you upload the step-by-step protocols used in this manuscript to the Protocol Exchange. Protocol Exchange is an open online resource that allows researchers to share their detailed experimental know-how. All uploaded protocols are made freely available, assigned DOIs for ease of citation and fully

searchable through nature.com. Protocols can be linked to any publications in which they are used and will be linked to from your article. You can also establish a dedicated page to collect all your lab Protocols. By uploading your Protocols to Protocol Exchange, you are enabling researchers to more readily reproduce or adapt the methodology you use, as well as increasing the visibility of your protocols and papers. Upload your Protocols at www.nature.com/protocolexchange/. Further information can be found at www.nature.com/protocolexchange/about.

Please note that *Nature Structural & Molecular Biology* is a Transformative Journal (TJ). Authors may publish their research with us through the traditional subscription access route or make their paper immediately open access through payment of an article-processing charge (APC). Authors will not be required to make a final decision about access to their article until it has been accepted. [Find out more about Transformative Journals](https://www.springernature.com/gp/open-research/transformative-journals)

Sincerely,

Dimitris Typas
Associate Editor
Nature Structural & Molecular Biology

ORCID: 0000-0002-8737-1319
